# Rectifying LLM Thought from Lens of Optimization

**Junnan Liu** [*]
Department of Data Science and AI, Faculty of Information Technology
Monash University
`junnan.liu@monash.edu`

**Hongwei Liu & Songyang Zhang[†] & Kai Chen[†]**
Shanghai Artificial Intelligence Laboratory
`{liuhongwei,zhangsongyang,chenkai}@pjlab.org.cn`

 **Code:** `https://github.com/open-compass/RePro`

## ABSTRACT

Recent advancements in large language models (LLMs) have been driven by their emergent reasoning capabilities, particularly through long chain-of-thought (CoT) prompting, which enables thorough exploration and deliberation. Despite these advances, long-CoT LLMs often exhibit suboptimal reasoning behaviors, such as overthinking and excessively protracted reasoning chains, which can impair performance. In this paper, we analyze reasoning processes through an optimization lens, framing CoT as a gradient descent procedure where each reasoning step constitutes an update toward problem resolution. Building on this perspective, we introduce REPRO (**Re**ctifying **Pro**cess-level Reward), a novel approach to refine LLM reasoning during post-training. REPRO defines a surrogate objective function to assess the optimization process underlying CoT, utilizing a dual scoring mechanism to quantify its intensity and stability. These scores are aggregated into a composite process-level reward, seamlessly integrated into reinforcement learning with verifiable rewards (RLVR) pipelines to optimize LLMs. Extensive experiments across multiple reinforcement learning algorithms and diverse LLMs, evaluated on benchmarks spanning mathematics, science, and coding, demonstrate that REPRO consistently enhances reasoning performance and mitigates suboptimal reasoning behaviors.

## 1 INTRODUCTION

Recent advancements in large language models (LLMs) have been propelled by their emergent reasoning capabilities, enabling them to tackle complex tasks (Huang & Chang, 2023; Plaat et al., 2024; Ahn et al., 2024; Ke et al., 2025; Sun et al., 2025). These capabilities are pivotal in progressing toward artificial general intelligence (AGI) (Zhong et al., 2024). State-of-the-art LLMs, such as OpenAI's o-series (OpenAI, 2024a;b; 2025), DeepSeek-R1 (DeepSeek-AI et al., 2025), Kimi-K1 (Kimi-Team et al., 2025), and Gemini-2.5-Pro (Comanici et al., 2025), leverage long chain-of-thought (CoT) prompting to enhance reasoning. This approach facilitates comprehensive exploration and reflection, yielding robust reasoning processes (Chen et al., 2025a). Such improvements stem largely from reinforcement learning with verifiable rewards (RLVR) (Schulman et al., 2017; Shao et al., 2024), which enables LLMs to autonomously explore reasoning steps based on a terminal reward, fostering self-improving models with scalable reasoning during inference (Snell et al., 2024).

Despite these advancements, long-CoT LLMs often exhibit suboptimal reasoning behaviors (Chen et al., 2025a). A significant issue is overthinking, where models generate excessive tokens or

---

[*]Work done when Junnan's internship at Shanghai Artificial Intelligence Laboratory. [†] Corresponding authors.

protracted reasoning paths that contribute minimally to problem resolution, incurring substantial computational costs (Chen et al., 2024; Wang et al., 2025c; Sui et al., 2025). For instance, in response to a simple query like "What is the answer to 2 plus 3?" (Chen et al., 2024), certain long-CoT LLMs produce reasoning chains exceeding thousands of tokens, increasing latency and resource demands, thus limiting applicability in time-sensitive domains (Sui et al., 2025).

Drawing on prior work (Feng et al., 2023; Huang et al., 2025a), we analyze suboptimal reasoning through an optimization framework, conceptualizing CoT as a task-specific variant of gradient descent, where each reasoning step represents an optimization update (Liu et al., 2025a). In this paradigm, suboptimal reasoning manifests as oscillations around saddle points or local optima, hindering convergence to the optimal solution.

To address these challenges, we propose REPRO (**Re**ctifying **Pro**cess-level Reward), a novel method to rectify LLM thought during post-training. REPRO formulates a surrogate objective function, $\mathcal{J}$, to monitor the optimization process of CoT, measuring the LLM's confidence in the ground truth via perplexity (Jelinek et al., 1977) over the ground-truth token sequence. For a reasoning trajectory of $N$ steps, we compute a sequence of objective values $[\mathcal{J}_0, \mathcal{J}_1, \ldots, \mathcal{J}_N]$ and introduce a dual scoring system to assess optimization intensity and stability. These scores are combined into a composite process-level reward (Lightman et al., 2024), integrated into standard post-training pipelines (DeepSeek-AI et al., 2025; Shao et al., 2024; Hu, 2025) to enhance reasoning. REPRO is plug-and-play, compatible with prevalent reinforcement learning algorithms.

The efficacy of REPRO is substantiated by comprehensive empirical evaluation. We validate RE-PRO through extensive experiments using reinforcement learning algorithms like PPO (Schulman et al., 2017), REINFORCE++ (Hu, 2025), REINFORCE++ Baseline (Hu, 2025), and GRPO (Shao et al., 2024), across LLMs of various families and scales, including base models, supervised fine-tuned variants, and native long-CoT LLMs. Evaluations on benchmarks in mathematics, science, and coding demonstrate significant improvements in reasoning performance. Quantitative and qualitative analyses further confirm REPRO's efficacy in optimizing reasoning behaviors. Our contributions are: ❶ We introduce REPRO, a plug-and-play method to rectify LLM reasoning in RLVR; ❷ We define a surrogate objective function to model reasoning as gradient descent, with a dual scoring mechanism for optimization intensity and stability, and outline its integration as a process-level reward; ❸ Extensive experiments across reinforcement learning algorithms and LLMs show enhanced reasoning performance; ❹ Quantitative and qualitative analyses verify REPRO's ability to refine LLM reasoning behaviors.

## 2 PRELIMINARIES

**Reinforcement Learning for LLM Reasoning.** Proximal Policy Optimization (PPO) (Schulman et al., 2017) is the typical and effective policy gradient algorithm for LLM post-training (Ouyang et al., 2022; Hu et al., 2025). As an actor-critic method, PPO employs a policy model (*actor*) to optimize a reward function and a value model (*critic*) to estimate the value of each state. PPO employs the clipped surrogate objective function to enhance training stability by constraining the magnitude of policy updates at each iteration with a clipping range $\epsilon$. Given the input data distribution $P$ and policy model $\pi_\theta$, the objective is formally defined as:

$$\mathcal{J}(\theta) = \mathbb{E}_{q \sim P, \boldsymbol{\tau} \sim \pi_\theta} \left[ \frac{1}{|\boldsymbol{\tau}|} \sum_{t=1}^{|\boldsymbol{\tau}|} \left\{ \min \left( \rho_t A_t, \, \text{clip}(\rho_t, 1 - \epsilon, 1 + \epsilon) \, A_t \right) \right\} \right], \quad (1)$$

where $\rho_t = \pi_\theta \left( \boldsymbol{\tau}_{(t)} | q, \boldsymbol{\tau}_{(\leq t)} \right) / \pi_{\theta_{\text{old}}} \left( \boldsymbol{\tau}_{(t)} | q, \boldsymbol{\tau}_{(\leq t)} \right)$ is the importance sampling coefficient to reduce the gap between the current policy and the old policy. $A_t$ denotes the advantage estimate at time step $t$, which is computed using Generalized Advantage Estimation (GAE) (Schulman et al., 2016). GAE is derived from the temporal difference error, $\delta_t = r_t + \gamma V_{t+1} = V_t$, where $r_t$ is the reward at time step $t$, $\gamma$ is the discount factor, and $V_t$ is the value at time step $t$. Then $A_t$ is calculated by the summation of the temporal difference error over a series of time steps as: $A_t = \sum_{i=0}^{\infty} \gamma^i \delta_{t+i}$.

**Critic-Free RL Algorithms for LLM Reasoning.** Despite the effectiveness of PPO, it experiences high computational costs due to the trainable value model. To address this challenge, a series of critic-free RL algorithms have been proposed, substituting the value $V_t$ with an estimated reward

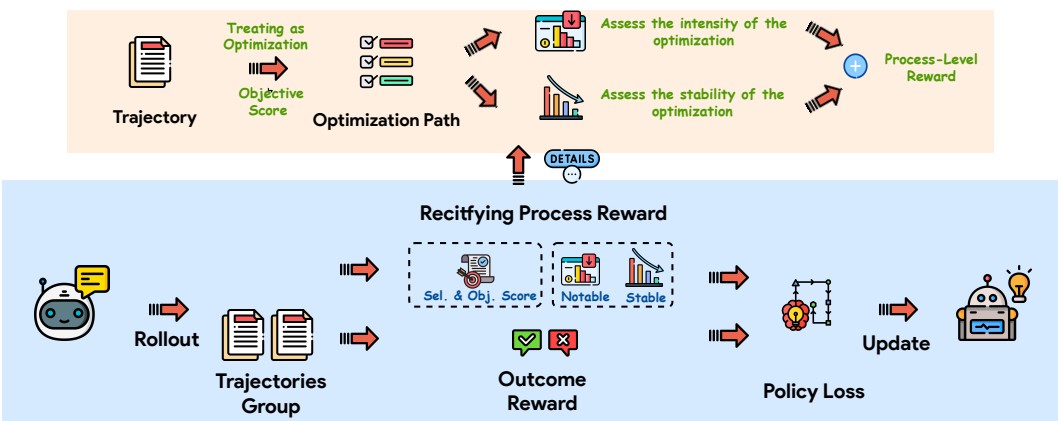

Figure 1: Illustration of the REPRO framework. We incorporate a rectifying process-level reward into the RLVR training to enhance LLM reasoning. Initially, we conceptualize the reasoning trajectories generated by LLMs as an optimization process of the LLMs' internal state (§ 3.1 & § 3.2). We then propose a two-fold score to evaluate the optimization process and utilize this score as a reward to rectify the LLM thought (§ 3.3 & § 3.4).

baseline. These include ReMax (Li et al., 2024), RLOO (Ahmadian et al., 2024), GRPO (DeepSeek-AI et al., 2025; Shao et al., 2024), and REINFORCE++ (Hu, 2025). Typically, these algorithms share the following objective function:

$$\mathcal{J}(\theta) = \mathbb{E}_{q \sim P, \{\boldsymbol{\tau}_i\} \sim \pi_\theta} \left[ \frac{1}{G} \sum_{i=1}^{G} \frac{1}{|\boldsymbol{\tau}_i|} \sum_{t=1}^{|\boldsymbol{\tau}_i|} \left\{ \min\left(\rho_{i,t} \tilde{A}_t^i, \operatorname{clip}(\rho_{i,t}, 1-\epsilon, 1+\epsilon) \, \tilde{A}_t^i\right) - \beta \, D_{\mathrm{KL}}\left[\pi_\theta \| \pi_{\mathrm{ref}}\right] \right\} \right],$$

$$(2)$$

where $\boldsymbol{\tau}_i = \{\boldsymbol{\tau}_1, \ldots, \boldsymbol{\tau}_G\} \sim \pi_{\theta_{\mathrm{old}}}(\cdot|q)$ denotes a group of trajectories of size $G$ generated by the existing policy model $\pi_\theta$. $\tilde{A}t^i$ represents the normalized advantage using an estimated reward baseline at time step $t$ for the $i$-th trajectory. $D_{\mathrm{KL}}\left[\pi_\theta|\pi_{\mathrm{ref}}\right]$ denotes the KL divergence penalty between the current policy $\pi_\theta$ and the reference policy $\pi_{\mathrm{ref}}$, with $\beta$ as the weighting factor for this penalty term.

# 3 REPRO: RECTIFYING LLM THOUGHT

In this section, we provide the details of the proposed REPRO and the illustration of REPRO is demonstrated in Figure 1.

## 3.1 PROBLEM FORMULATION

A typical LLM reasoning process involves a question $\boldsymbol{q}$ randomly sampled from the question distribution $P(Q)$, denoted as $\boldsymbol{q} \sim P(Q)$, and an LLM parameterized by $\pi_\theta$. For $\boldsymbol{q}$, a long-CoT LLM generates a step-by-step reasoning sequence $\boldsymbol{\tau}_{\mathrm{thinking}}$ (typically delimited by `<think>` and `</think>` tags in current reasoning LLMs), followed by a conclusion $\boldsymbol{\tau}_{\mathrm{conclusion}}$, forming the trajectory:

$$\boldsymbol{\tau} = [\boldsymbol{\tau}_{\mathrm{thinking}}; \boldsymbol{\tau}_{\mathrm{conclusion}}] \sim \pi_\theta(\cdot|\boldsymbol{q}). \qquad (3)$$

Following prior work (Liu et al., 2025a; Wang et al., 2025a), we conceptualize the decoding of $\boldsymbol{\tau}_{\mathrm{thinking}}$ as an optimization process over the LLM's internal states, iteratively increasing the likelihood of the correct answer. The objective function $\mathcal{J}(\pi_\theta, \boldsymbol{q}, \boldsymbol{\tau}, \boldsymbol{a})$, where $\boldsymbol{a}$ is the ground-truth answer, is optimized as:

$$\theta_{t+1} \leftarrow \theta_t + \tilde{\eta} \cdot \tilde{\nabla}_\theta \mathcal{J}(\pi_\theta, \boldsymbol{q}, \boldsymbol{\tau}_{(\leq t)}, \boldsymbol{a}), \quad \theta^* = \arg\max_\theta \mathcal{J}(\pi_\theta, \boldsymbol{q}, \boldsymbol{\tau}, \boldsymbol{a}), \qquad (4)$$

where $\tilde{\eta}$ is an implicit learning rate, and $\tilde{\nabla}_\theta \mathcal{J}(\pi_\theta, \boldsymbol{q}, \boldsymbol{\tau}_{(\leq t)}, \boldsymbol{a})$ denotes the implicit gradient of $\mathcal{J}(\pi_\theta, \boldsymbol{q}, \boldsymbol{\tau}_{(\leq t)}, \boldsymbol{a})$ with respect to $\theta$, as the actual optimization process is complex and nontrivial.

## 3.2 OBJECTIVE FUNCTION DEFINITION

Although the actual optimization process is complex and nontrivial, we can define a proxy metric to observe changes in the objective function from an indirect perspective. Drawing inspiration from previous work (Tang et al., 2025; Zhou et al., 2025; Yu et al., 2025b), we find that the probability of the model generating the ground truth answer $\boldsymbol{a}$ serves as an effective proxy for the objective function $\mathcal{J}$. Formally, we define the proxy objective function as follows:

$$\tilde{\mathcal{J}}\left(\pi_\theta, \boldsymbol{q}, \boldsymbol{\tau}_{(\leq t)}, \boldsymbol{a}\right) \triangleq \frac{1}{|\boldsymbol{a}|} \sum_{i=1}^{|\boldsymbol{a}|} \log \pi_\theta\left(\boldsymbol{a}_{(i)}|\boldsymbol{q}, \boldsymbol{\tau}_{(\leq t)}\right). \tag{5}$$

Intuitively, $\tilde{\mathcal{J}}$ quantifies the model's reasoning capability given certain context. As $\boldsymbol{\tau}_{\leq t}$ updates the model's internal states, the probability of producing the ground-truth answer increases, thereby increasing $\tilde{\mathcal{J}}$.

**Empirical Evidence.** We provide empirical evidence supporting the effectiveness of $\tilde{\mathcal{J}}$ as a proxy metric. Specifically, we prompt LRM (DeepSeek-R1-Distill-Qwen-1.5B) (DeepSeek-AI et al., 2025) with a mathematical question sampled from AIME'24[1] to generate multiple reasoning trajectories. We select four correct trajectories, computing $\tilde{\mathcal{J}}$ (to demonstrate, we show the negative value of $\tilde{\mathcal{J}}$) at each position of the trajectory, and plot the curve as shown in Figure 2. From Figure 2, we observe that $-\tilde{\mathcal{J}}$ gradually decreases as the reasoning trajectory length increases.

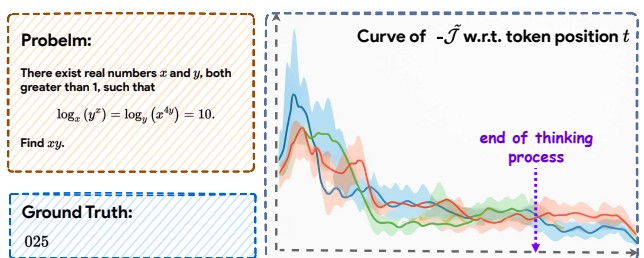

Figure 2: Empirical evidence supporting $-\tilde{\mathcal{J}}$ as a proxy metric. The left panel presents the question and its corresponding answer, while the right panel plots $-\tilde{\mathcal{J}}$ as a function of reasoning trajectory tokens.

This indicates that $\tilde{\mathcal{J}}$ effectively serves as a proxy metric for monitoring and assessing the internal states of the LLM.

## 3.3 QUANTIFYING OPTIMIZATION PROCESS

Leveraging the proposed objective function $\tilde{\mathcal{J}}$, we introduce a score $\mathcal{S}$ designed to evaluate the optimization process by tracking the dynamics of $\tilde{\mathcal{J}}$. For a given reasoning trajectory $\boldsymbol{\tau}$, a sequence of $\tilde{\mathcal{J}}$ values, denoted as $\{\tilde{\mathcal{J}}1, \tilde{\mathcal{J}}_2, \ldots, \tilde{\mathcal{J}}|\boldsymbol{\tau}|\}$, is obtained, which represents the optimization process over the generation of $\boldsymbol{\tau}$. An effective optimization process should fulfill two key conditions: 1) the value of objective function exhibits a sufficient overall increase, indicating substantial progress to the optimization objective; 2) the increase is relatively smooth, with limited oscillation near local extrema, indicating efficient optimization. Building on these criteria, we propose a dual quantitative score, $\mathcal{S}$, to evaluate the optimization process. This score comprises the **Magnitude Score**, $\mathcal{S}_{\text{magn}}$ measuring the *intensity* of the optimization process (i.e., net improvement), and the **Stability Score**, $\mathcal{S}_{\text{stab}}$, assessing its *stability*, capturing the degree of oscillatory behavior in the updates.

**Magnitude Score.** The magnitude score, $\mathcal{S}_{\text{magn}}$, at position $t$ (denoted as $\mathcal{S}_{\text{magn},(t)}$), quantifies the increase in $\tilde{\mathcal{J}}$ along the partial trajectory $\boldsymbol{\tau}_{\leq t}$. To address the disparities among $\tilde{\mathcal{J}}$ values corresponding to different $\boldsymbol{q}$, a baseline $\overline{\mathcal{J}}_b(\boldsymbol{q})$ is introduced, defined as:

$$\overline{\mathcal{J}}_b(\boldsymbol{q}) \triangleq \tilde{\mathcal{J}}\left(\pi_\theta, \boldsymbol{q}, \boldsymbol{a}\right), \tag{6}$$

---

[1] https://huggingface.co/datasets/HuggingFaceH4/aime_2024

which can be interpreted as the direct probabilistic prediction from $\pi_\theta$. Subsequently, $\mathcal{S}_{\text{magn},(t)}$ is defined as:

$$\mathcal{S}_{\text{magn},(t)} \triangleq \tanh\left(\Delta(\pi_\theta, \boldsymbol{q}, \boldsymbol{\tau}_{(\leq t)}, \boldsymbol{a}) + 1\right) + 1 \in (0, 1],$$

$$\text{where } \Delta(\pi_\theta, \boldsymbol{q}, \boldsymbol{\tau}_{(\leq t)}, \boldsymbol{a}) = \frac{\tilde{\mathcal{J}}\left(\pi_\theta, \boldsymbol{q}, \boldsymbol{\tau}_{(\leq t)}, \boldsymbol{a}\right) - \overline{\mathcal{J}}_b(\boldsymbol{q})}{\overline{\mathcal{J}}_b(\boldsymbol{q})} \tag{7}$$

Intuitively, $\mathcal{S}_{\text{magn},(t)}$ is a normalized measure of the relative increase of $\tilde{\mathcal{J}}$ over the baseline $\overline{\mathcal{J}}_b$. Normalization of this relative decrease $\Delta(\pi_\theta, \boldsymbol{q}, \boldsymbol{\tau}_{\leq t}, \boldsymbol{a})$ through the tanh function ensures the score's range is restricted to $(0, 1]$, thus mitigating the impact of extreme values. A higher $\mathcal{S}_{\text{magn},(t)}$ signifies a greater increase in the objective function, concurrently indicating that the partial reasoning trajectory $\boldsymbol{\tau}_{\leq t}$ yields more substantial benefits to the reasoning process.

**Stability Score.** As previously stated, the $\mathcal{S}_{\text{stab}}$ quantifies the stability of the optimization process. Each step is expected to serve as an effective update, progressing towards increasing the objective function. For a given sequence of objective values $\{\tilde{\mathcal{J}}_1, \tilde{\mathcal{J}}_2, \ldots, \tilde{\mathcal{J}}_{|\boldsymbol{\tau}_{(\leq t)}|}\}$, we evaluate $\mathcal{S}_{\text{stab},(t)}$ by examining its correlation with the corresponding indices $\{1, 2, \ldots, |\boldsymbol{\tau}_{(\leq t)}|\}$, as follows:

$$\mathcal{S}_{\text{stab},(t)} = \frac{\sum_{i<j, 1\leq i,j\leq |\boldsymbol{\tau}_{(\leq t)}|} \text{sign}\left(\tilde{\mathcal{J}}_i - \tilde{\mathcal{J}}_j\right) \cdot \text{sign}\left(i - j\right)}{|\boldsymbol{\tau}_{(\leq t)}|\left(|\boldsymbol{\tau}_{(\leq t)}| - 1\right)} + \frac{1}{2} \in [0, 1], \tag{8}$$

where sign is the sign function. Theoretically, we leverage Kendall's Tau Correlation Coefficient (Kendall, 1938) to measure the stability of the optimization process. If each step is an effective update, $\mathcal{S}_{\text{stab}}$ approaches 1, whereas ineffective updates result in a score near 0. In addition, considering the influence of noise, smooth can also be introduced to smooth $\tilde{\mathcal{J}}_i$, such as the common EMA (Exponential Moving Average) (Hunter, 1986) smoothing equation:

$$\tilde{\mathcal{J}}_{i,t} = \alpha \cdot \tilde{\mathcal{J}}_{i,t-1} + (1 - \alpha) \cdot \tilde{\mathcal{J}}_{i,t}, \tag{9}$$

where $\alpha$ is the smoothing factor.

Finally, combining $\mathcal{S}_{\text{magn}}$ and $\mathcal{S}_{\text{stab}}$, we obtain the final score $\mathcal{S}$ by introducing a weight factor $w \in [0, 1]$ as follows:

$$\mathcal{S} = (1 - w) \cdot \mathcal{S}_{\text{magn}} + w \cdot \mathcal{S}_{\text{stab}}. \tag{10}$$

### 3.4 LEARNING WITH RECTIFYING PROCESS-LEVEL REWARD

We shall now discuss the integration of $\mathcal{S}$ into the RL training for LLMs. A natural approach is to employ $\mathcal{S}$ as a process reward in reinforcement learning training. Nonetheless, since current strong reasoning LLMs generate lengthy reasoning trajectories, computing $\mathcal{S}$ at each token would incur prohibitive computational overhead. Moreover, token-level $\mathcal{S}$ calculation could introduce excessive noise, leading to futile computations of $\mathcal{S}$ values that adversely affect the optimization process.

**Entropy-Based Selection Strategy.** Recent studies (Cui et al., 2025; Wang et al., 2025b; Cheng et al., 2025) have demonstrated that token entropy serves as an effective indicator within the trajectories of advanced reasoning LLMs. High-entropy tokens act as critical decision points, guiding the model toward diverse reasoning pathways. In this study, where reasoning trajectory generation is framed as an optimization process, high-entropy tokens may cause oscillations near extrema, yielding higher value than tokens with lower entropy. Therefore, to reduce computational overhead while providing more effective process rewards, we propose an entropy-based selection strategy. Specifically, we divide the thinking tokens within the reasoning trajectory into multiple segments (e.g., partitioned by two-line-break \n\n), considering the thinking granularity of LLMs:

$$\boldsymbol{\tau}_{\text{thinking}} \mapsto \{\boldsymbol{c}_1, \boldsymbol{c}_2, \ldots, \boldsymbol{c}_N\}, \tag{11}$$

where $N$ denotes the number of segments. We select the top-$k$ segments $\{\tilde{c}_1, \tilde{c}_2, \ldots, \tilde{c}_k\}$ based on the entropy of the first token of each segment:

$$\text{top-}k\left(\mathcal{H}\left(\boldsymbol{c}_{1,(0)}\right), \mathcal{H}\left(\boldsymbol{c}_{2,(0)}\right), \ldots, \mathcal{H}\left(\boldsymbol{c}_{N,(0)}\right)\right) \mapsto \{\tilde{\boldsymbol{c}}_1, \tilde{\boldsymbol{c}}_2, \ldots \tilde{\boldsymbol{c}}_k\}. \tag{12}$$

The rationale for this strategy is that model uncertainty increases at the conclusion of these segments, which, from an optimization perspective, indicates that suboptimal optimization processes are more likely to occur, a phenomenon we seek to rectify.

**Integrating $\mathcal{S}$ Into Reward.** Given the selected segments $\{\tilde{c}_1, \tilde{c}_2, \ldots, \tilde{c}_k\}$, we compute scores $\{\mathcal{S}_1, \mathcal{S}_2, \ldots, \mathcal{S}_k\}$ for each segment by appending the ground truth answer at the end. The rectifying process-level reward $\tilde{r}_j$ is computed as:

$$\tilde{r}_j = \begin{cases} \mathcal{S}_j - \mathcal{S}_{j-1} & \text{if } j > 1, \\ \mathcal{S}_j & \text{if } j = 1, \end{cases} \tag{13}$$

which signifies the gain from introducing the partial trajectory from the end of $\tilde{c}_{j-1}$ to $\tilde{c}_j$. $\tilde{r}_j$ rectifies the thinking process of LLMs by penalizing thinking processes associated with suboptimal optimization while encouraging those associated with optimal optimization. Subsequently, we also involve the normalization in critic-free RL algorithms on the rectifying process-level reward $\tilde{r}_t$ to mitigate the probabilistic prediction mismatch between different $q$ and facilitate stable policy updates:

$$\tilde{r}_j' = \text{Norm}(\tilde{r}_j | \{\tilde{r}_{j,i}\}_i), \tag{14}$$

where $\{\tilde{r}_{j,i}\}_i$ denotes the specific group for the normalization of $r_j'$ and refer to Appendix B.1 for more details. We separate the normalization of process-level reward from outcome reward to prevent interference with the correctness reward from noise signals. For a token $\tau_{(t)}$ where $\text{index}\left(c_{j-1,(|c_{j-1}|)}\right) < t \leq \text{index}\left(c_{j,(0)}\right)$ and $\text{index}(x)$ denotes the index of token $x$, the rectifying process-level advantage $\tilde{A}_t$ and overall advantage $\hat{A}_t$ are defined as $\tilde{A}_t = \sum_{i=j}^{k} \tilde{r}_i', \hat{A}_t = A + \alpha \cdot \tilde{A}_t$ where $A$ is the verifiable outcome advantage to measure the correctness of the overall trajectory, computed from the correctness of reasoning trajectories, and $\alpha$ is the weight factor.

By synthesizing all propositions and definitions outlined above, the LLM parameters are updated using the computed advantage $\hat{A}_t$ following the clipped surrogate objective specified in Equation (2).

**Empirical Example For Understanding $\tilde{r}$.** To illustrate the effectiveness of the rectifying process-level reward $\tilde{r}$, we present an empirical example. As shown in Example 3.1, we prompt the LRM to answer a question, compute $\tilde{r}$ for each reasoning step, and identify steps with low rectifying process-level rewards. We find that steps with low $\tilde{r}$ typically involve self-doubt or redundant re-examination, contributing little to the reasoning process and increasing inefficiency. Conversely, steps with high $\tilde{r}$ correspond to critical computational actions or preliminary conclusions that significantly advance reasoning. Thus, we propose that penalizing low-$\tilde{r}$ steps and incentivizing high-$\tilde{r}$ steps can optimize the model's reasoning behavior.

---

**Example 3.1: Empirical Example for Rectifying Process-Level Reward**

QUESTION:
There exist real numbers $x$ and $y$, both greater than 1, such that

$$\log_x (y^x) = \log_y (x^{4y}) = 10.$$

Find $xy$.

SELECTED REASONING PROCESS WITH HIGH/LOW RECTIFYING PROCESS-LEVEL REWARD:

($\tilde{r} = 0.143$) *Alright, so I have this problem here: There exist real numbers $x$ and $y$, both greater than 1. ... ($\tilde{r} = -0.217$) Hmm, okay. Let's see. I remember that logarithms can be tricky, but maybe I can use some logarithmic identities to simplify these equations. I think the change of base formula might be helpful here. The change of base formula says that $\log_b a = \frac{\ln a}{\ln b}$, right? So maybe I can rewrite both equations using natural logarithms to make them easier to handle. ... ($\tilde{r} = -0.053$) That seems a bit cleaner. So, let me write those. ... ($\tilde{r} = -0.060$) This trial and error method isn't working. Maybe I should consider a different approach. ... ($\tilde{r} = 0.085$) Left side: $1(1 + 10e^{-1}) \approx 1 + 10 \times 0.3679 \approx 1 + 3.679 \approx 4.679$, which is greater than 3.2188. ... ($\tilde{r} = -0.113$) Wait, that can't be. Wait, perhaps my approximation is off. ... Wait, perhaps my approach is not efficient. ($\tilde{r} = -0.086$) Maybe I can use linear approximation or try to set up an equation. ... ($\tilde{r} = 0.053$) Which is the same equation as before. So, this equation is satisfied when $xy = 25$. So, $xy = 25$ is the solution. ... ($\tilde{r} = 0.092$) Therefore, despite the complexity of the original logarithmic equations, the product $xy$ simplifies directly to 25.*

---

Table 1: Performance of REPRO on evaluation reasoning benchmarks. We report the average performance for 16 runs on AIME24 and AIME25, and 4 runs on others. We abbreviate LMB as LiveMathBench v202505, LCB as LiveCodeBench v6, RF++ as REINFORCE++, and RF++ B as REINFORCE++ Baseline. ♠ denotes the in-domain evaluation benchmark and ♣ denotes the out-of-domain benchmark.

| Methods | AIME24 ♠ % Avg@16 ↑ | AIME25 ♠ % Avg@16 ↑ | MATH500 ♠ % Avg@4 ↑ | LMB ♠ % Avg@4 ↑ | GPQA-D ♣ % Avg@4 ↑ | MBPP ♣ % Avg@4 ↑ | LCB ♣ % Avg@4 ↑ |
|---|---|---|---|---|---|---|---|
| *DeepSeek-R1-Distill-Qwen-1.5B* | | | | | | | |
| Original | 30.6 | 24.8 | 84.4 | 10.5 | 32.7 | 61.8 | 14.9 |
| PPO | 34.8 | 24.4 | 86.9 | 14.0 | 32.1 | 61.0 | 17.0 |
| +REPRO | 36.3 | 27.7 | 87.7 | 16.5 | 32.8 | 61.1 | 16.7 |
| RF++ | 31.0 | 23.5 | 85.4 | 11.5 | 33.1 | 61.0 | 15.1 |
| +REPRO | 33.1 | 26.7 | 86.1 | 12.0 | 34.3 | 61.9 | 16.1 |
| RF++ B | 33.1 | 26.7 | 86.1 | 12.0 | 34.3 | 61.9 | 16.1 |
| +REPRO | 35.6 | 26.5 | 87.2 | 15.6 | 35.4 | 63.9 | 17.3 |
| GRPO | 32.9 | 25.3 | 86.0 | 10.3 | 34.5 | 62.5 | 15.2 |
| +REPRO | 36.0 | 26.5 | 87.1 | 14.3 | 37.0 | 65.4 | 18.4 |
| *Qwen3-1.7B* | | | | | | | |
| Original | 46.8 | 36.1 | 93.0 | 18.8 | 39.5 | 66.9 | 30.6 |
| GRPO | 47.3 | 34.8 | 93.4 | 18.8 | 38.3 | 67.5 | 32.2 |
| +REPRO | 49.8 | 37.9 | 94.1 | 19.5 | 39.1 | 68.8 | 32.0 |

## 4 EXPERIMENTAL RESULTS AND ANALYSIS

### 4.1 SETUP

**Evaluation Benchmarks.** We evaluate all models across three domain benchmarks: 1) *Mathematical Reasoning Benchmarks*, which include AIME24, AIME25, and MATH500 (Hendrycks et al., 2021b), and LiveMathBench (Liu et al., 2024); 2) *Scientific Reasoning Benchmarks*, represented by GPQA-Diamond (Rein et al., 2023); 3) *Code Reasoning Benchmarks*, which comprise MBPP (Austin et al., 2021) and LiveCodeBench (Jain et al., 2025).

**Implementation Details.** We conduct experiments on several prominent LLMs, including DeepSeek-R1-Distill-Qwen-1.5B (DeepSeek-AI et al., 2025), distilled from DeepSeek-R1 (DeepSeek-AI et al., 2025), Qwen3-1.7B (Yang et al., 2025), Qwen3-8B (Yang et al., 2025), Hunyuan-1.8B-Instruct (Tencent, 2025), and MobileLLM-R1-950M (Liu et al., 2025c). The training corpus, proposed by Luo et al. (2025), comprises approximately 40,000 high-quality mathematical samples. More details of the model training are provided in Appendix B.4. For evaluation, we configured the sampling temperature to 0.6, top-$p$ to 0.95, and top-$k$ to 40. It is worth mentioning that different LLMs have their recommended sampling parameters, and different sampling parameters may have a certain impact on performance. However, to ensure a consistent evaluation pipeline, we use same sampling parameters for all models and focus on the *relative* improvement in performance. Also, to minimize variance, we report average performance relative to the size of each benchmark.

### 4.2 EFFECTIVENESS AND GENERALIZATION OF REPRO

Table 1 illustrates the performance of REPRO on evaluation reasoning benchmarks. From the experimental results, we have the following findings.

**REPRO Achieves Consistent Improvements Across RL Algorithms.** As presented in Table 1, RE-PRO consistently enhances the performance of various RL baselines, including PPO, REINFORCE++, REINFORCE++ Baseline, and GRPO. For instance, when applied to PPO on the DeepSeek-R1-Distill-Qwen-1.5B backbone, REPRO improves AIME24 accuracy from 34.8 to 36.3 and MATH500 from 86.9 to 87.7. These results demonstrate that REPRO's improvements are independent of the RL algorithm, offering a versatile, plug-and-play strategy to enhance reasoning performance.

**REPRO Generalizes to Out-of-Domain Benchmarks.** Beyond in-domain reasoning tasks like AIME and MATH500, REPRO exhibits robust generalization to out-of-domain benchmarks, such as

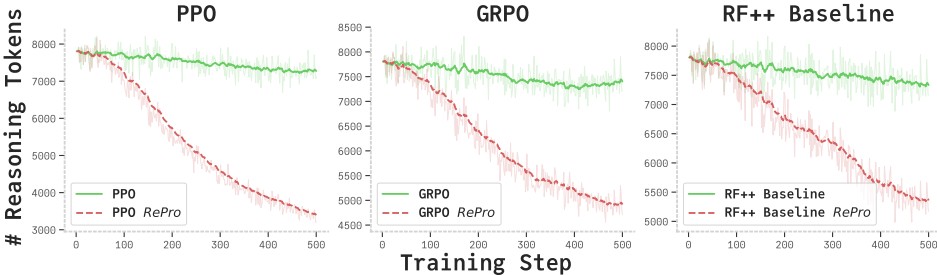

Figure 3: Dynamics of the reasoning token cost during the training process of REPRO on DeepSeek-R1-Distill-Qwen-1.5B.

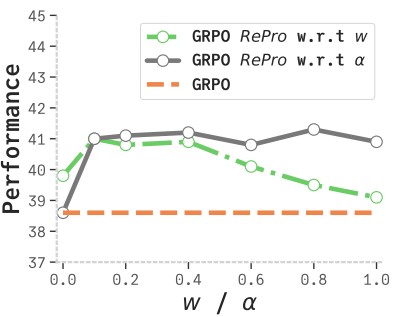

Figure 4: Ablation experiments of weight $w$ and REPRO weight $\alpha$.

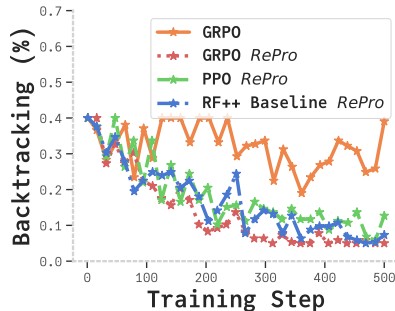

Figure 5: Dynamics of the *backtracking* pattern.

science reasoning benchmark GPQA-Diamond and code reasoning benchmarks including MBPP and LiveCodeBench. These findings underscore REPRO's ability to extend benefits beyond mathematical reasoning to diverse reasoning tasks like science reasoning, programming, and code generation tasks, highlighting its broad applicability.

**REPRO Generalizes to Diverse LLMs.** REPRO's effectiveness extends across different LLMs including DeepSeek-R1-Distill-Qwen-1.5B and Qwen3-1.7B. We also provide results of different architectures and sizes in Appendices C.1 and C.2. REPRO achieves consistent improvements across LLMs of different families and sizes. This scalability across model architectures and sizes indicates that REPRO serves as a general mechanism for enhancing reasoning capabilities, rather than an optimization specific to a particular backbone.

### 4.3 ABLATION STUDY

**Impact of Magnitude Score & Stability Score.** Magnitude Score and Stability Score are utilized to assess the intensity and stability of the optimization process, respectively, with the coefficient $w$ balancing their contributions. To investigate their necessity, we analyze model performance across different $w$ values, as shown in Figure 4. The dotted lines, segmented by dots, represent the average performance of models trained with varying $w$ values on four mathematical reasoning benchmarks. Across all $w$ values, REPRO consistently outperforms the baseline, confirming the importance of both Magnitude Score and Stability Score. Notably, performance is slightly higher when $w$ is lower, suggesting that the Magnitude Score, which measures optimization intensity, may play a more critical role in enhancing model performance.

**Impact of the REPRO Weight $\alpha$.** To balance the outcome advantage and REPRO advantage, we introduce the coefficient $\alpha$, set to 0.1. In this part, we evaluate the sensitivity of REPRO to variations in $\alpha$. As depicted in Figure 4, the solid lines, segmented by dots, represent the average performance of models trained with different $\alpha$ values across four mathematical reasoning benchmarks. The results

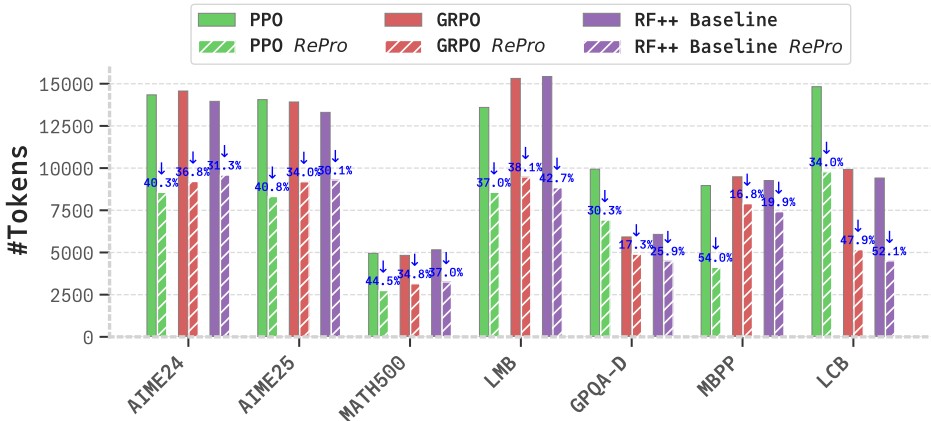

Figure 6: Comparison of the inference reasoning token cost of DeepSeek-R1-Distill-Qwen-1.5B.

Table 2: Ablation study of the number of selected segments $k$.

| # $k$ | AIME24 ♠ % Avg@16 ↑ | AIME25 ♠ % Avg@16 ↑ | MATH500 ♠ % Avg@4 ↑ | LMB ♠ % Avg@4 ↑ | GPQA-D ♣ % Avg@4 ↑ | MBPP ♣ % Avg@4 ↑ | LCB ♣ % Avg@4 ↑ |
|---|---|---|---|---|---|---|---|
| | | | *DeepSeek-R1-Distill-Qwen-1.5B* | | | | |
| 10 | 36.0 | 26.5 | 87.1 | 14.3 | 37.0 | 55.4 | 18.4 |
| 20 | 36.5 | 26.2 | 87.3 | 14.7 | 37.5 | 55.0 | 19.1 |
| 30 | 36.9 | 27.2 | 87.8 | 15.1 | 37.6 | 55.3 | 19.5 |

demonstrate that REPRO maintains relatively stable performance across various $\alpha$ values, indicating its robustness to changes in the balance coefficient.

**Impact of the Number of Selected Segments $k$.** The number of selected segments $k$ is a key factor in balancing performance and training cost in REPRO. As shown in Table 2, while increasing $k$ yields slight performance improvements, these gains are marginal. In practical applications, finding an optimal trade-off between training cost and performance is crucial. Additional experimental results and analysis are provided in Appendix C.4.

## 4.4 FURTHER ANALYSIS OF THE CHANGING OF THINKING BEHAVIORS

In this section, we will conduct quantitative and qualitative analyses of reasoning behaviors beyond reasoning performance to demonstrate how REPRO has improved the LLM thought.

**REPRO Improves the Token Efficiency of Reasoning.** We analyze the token cost of REPRO, examining its impact on reasoning efficiency. Figure 3 illustrates the dynamic changes in reasoning token cost during training with and without REPRO. The results show that REPRO effectively reduces token cost as training progresses. Additionally, we compare REPRO's token cost against baselines in Figure Figure 6, demonstrating significant reductions in inference token cost across all benchmarks. These findings indicate that REPRO promotes more concise and effective reasoning trajectories, enhancing both efficiency and practical applicability.

**REPRO Reduces Suboptimal Thinking Behaviors.** To gain deeper insight into the impact of REPRO beyond mere token counts, we analyze changes in the thinking patterns of LLMs. Every 10 steps during the REPRO training process, we instruct the LLM to perform reasoning on the AIME24 benchmark and apply the prompt proposed by Gandhi et al. (2025) (detailed in Appendix B.5) for thinking pattern recognition. Figure 5 illustrates the evolving proportion of the *backtracking* pattern, which typically indicates ineffective and excessive reasoning. As training progresses, the prevalence of this suboptimal pattern significantly decreases compared with vanilla GRPO, highlighting REPRO's effectiveness in enhancing the reasoning behaviors and patterns of LLMs.

**Case Study.** We also present a qualitative analysis of the responses generated by REPRO, as illustrated in Case D.1, Case D.2, and Case D.3. Compared to responses from the LLM trained with a vanilla RL algorithm, REPRO significantly reduces inefficient and suboptimal backtracking and reasoning (highlighted in orange), resulting in a more linear and efficient thinking process. Furthermore, by mitigating oscillations around "saddle points", REPRO-trained models exhibit fewer errors (marked in red), enhancing overall reasoning accuracy.

## 5 RELATED WORK

**Demystifying Reasoning Trajectories of LLMs.** The advent of powerful reasoning LLMs, enhanced by reasoning trajectories, has spurred extensive research to uncover the underlying mechanisms within these trajectories. Foundational studies (Yun et al., 2020a;b) have established that sufficiently expressive Transformers (Vaswani et al., 2017) can act as universal approximators for continuous sequence-to-sequence mappings over compact domains. Subsequent analyses have explored their computational power and expressive limitations (Dehghani et al., 2019; Bhattamishra et al., 2020; Yao et al., 2021; Hewitt et al., 2020; Weiss et al., 2021; Merrill et al., 2022; Chiang et al., 2023; Giannou et al., 2023; Liu et al., 2023). Recent research has shown that Transformers are capable of meta-learning optimization algorithms, such as gradient descent, within their forward trajectories (Gatmiry et al., 2024; Huang et al., 2025b). The most related works (Dai et al., 2023; Liu et al., 2025a) treat the reasoning trajectories of LLMs as optimization processes for their parameters and internal states, providing a solid foundation for our work.

**Promoting and Improving LLM Reasoning.** RL has emerged as a powerful paradigm for enhancing the reasoning capabilities of LLMs, with a notable approach being RLVR (OpenAI, 2024a;b; 2025; DeepSeek-AI et al., 2025; Kimi-Team et al., 2025; Team, 2025; Yang et al., 2025; Comanici et al., 2025; ByteDance Seed, 2025). Many of these methods leverage test-time scaling, a process where models engage in iterative self-improvement by refining their internal thought processes, exploring diverse strategies, and executing self-correction, often guided by CoT prompting. The resulting models, often termed long-CoT LLMs, have shown remarkable performance improvements on complex tasks in domains like mathematics, science, and code. More recent work has focused on refining the RL algorithms themselves. For instance, methods such as Dr.GRPO (Liu et al., 2025d), VAPO (Yue et al., 2025), and DAPO (Yu et al., 2025a) introduce algorithmic adaptations, particularly in sampling strategies and advantage estimation, to further elevate the reasoning performance of LLMs. A significant limitation of long-CoT LLMs is their computational inefficiency, often resulting in "overthinking", characterized by the generation of redundant tokens or unnecessary reasoning steps that may lead to errors (Chen et al., 2024; Liu et al., 2024; Wang et al., 2025c). To address this issue, one research direction (Hou et al., 2025; Aggarwal & Welleck, 2025; Liu et al., 2025b; Wu et al., 2025b; Wang et al., 2025a) focuses on introducing new forms of regularization or rewards based on length or information during training to reduce invalid token consumption. Another research approach (Yang et al., 2025; Fang et al., 2025; Wu et al., 2025a; Zhang et al., 2025) aims to learn adaptive policies that control the reasoning process by altering the reasoning pattern according to question difficulty or user instructions.

## 6 CONCLUSION

In this paper, we propose REPRO, a novel framework designed to refine the reasoning processes of LLMs from an optimization perspective. We conceptualize CoT reasoning as an optimization process and introduce two scores to evaluate its intensity and stability. These scores are integrated as a process-level reward into the training pipeline of RLVR. Extensive experiments across diverse reasoning benchmarks demonstrate the effectiveness of REPRO. Furthermore, we illustrate how REPRO enhances the reasoning behavior of LLMs, improving their efficiency.

ETHICS STATEMENT

This study focuses solely on general research tasks and poses no risks to health, safety, personal security, or privacy. No human participants are involved, and no new datasets are released as part of this work. Furthermore, the research does not include potentially harmful insights, methods, or applications, nor does it raise concerns related to privacy, security, legal compliance, or research integrity. Consequently, we anticipate no ethical risks or conflicts of interest. We are committed to maintaining the highest standards of scientific integrity and adhering to ethical guidelines throughout all stages of the research process.

REPRODUCIBILITY STATEMENT

We provide a comprehensive description of the proposed REPRO framework in § 3. To ensure reproducibility, we detail implementation specifics, including datasets, model configurations, and additional information, in § 4.1 and Appendix B. Key code implementations are included in the supplementary materials, with the complete code to be released publicly upon acceptance of the paper.

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

# Appendix

# Table of Contents

## A    DISCUSSIONS

**Computational Efficiency.** While REPRO necessitates additional computation due to the entropy-based selection strategy and reward calculation, these forward processes exhibit significant prefix overlap. Modern LLM inference engines like vLLM (Kwon et al., 2023) and SGLang (Zheng et al., 2024) expedite forward computation by caching key-value pairs (KV cache) (Shazeer, 2019). In our training process, we utilize the inference engines supported by the training framework to accelerate the computation of $\mathcal{S}$.

**Comparison with Process Reward Models.** Process reward models (Lightman et al., 2024; Wang et al., 2024) have been proposed to offer process-level granularity in supervising the training of reasoning LLMs. However, these methods struggle to generate effective supervision signals due to the complex dependencies among different processes and the final answers (DeepSeek-AI et al., 2025). In contrast, the method proposed in this paper, REPRO, neither relies on additional models nor provides absolute supervision signals of correctness or incorrectness. Instead, it evaluates the contribution of each process to reasoning and offers relative advantages and disadvantages through group normalization, significantly reducing noise in the signals provided for training.

**Comparison with Efficient Reasoning Methods.** Recent studies on efficient LLM reasoning (Aggarwal & Welleck, 2025; Hou et al., 2025; Liu et al., 2025b) emphasize the inclusion of a penalty coefficient for response length during the post-training process of LLMs, which rewards shorter, correct trajectories and penalizes longer ones. These methods only perform a coarse-grained length-based evaluation of the model's trajectories, which can easily lead to incorrect penalties for necessary and correct trajectories, thereby having a negative impact on the model's performance (Liu et al., 2025b). Alternatively, our method provides a more nuanced process reward, enhancing the LLMs' capability and efficiency by refining its reasoning patterns with finer granularity. As shown in Table 3, REPRO significantly outperforms salient baselines focusing on reasoning efficiency. Compared to the Vanilla GRPO baseline, REPRO achieves superior performance across all benchmarks (e.g., +3.1% on AIME24) while reducing token consumption by approximately 50% (12,367 to 6,158 tokens). In contrast to efficiency-focused methods like ThinkPrune and Laser, which suffer significant performance drops to achieve lower latency, REPRO successfully decouples efficiency from accuracy degradation. For instance, while L1-Max matches our token efficiency, it severely lags in reasoning capability (26.7% vs. 36.0% on AIME24), highlighting REPRO's ability to condense reasoning without information loss. Similarly, REPRO surpasses AdaThink by a clear margin of 4.3% on AIME24 while requiring 28% fewer tokens, demonstrating a more effective internal optimization than external stopping criteria.

Table 3: Comparison bewteen REPRO and ThinkPrune (Hou et al., 2025), L1 (Aggarwal & Welleck, 2025), Laser (Liu et al., 2025b), and AdaThink (Zhang et al., 2025).

| Methods | AIME24 % Avg@16 ↑ | AIME25 % Avg@16 ↑ | MATH500 % Avg@4 ↑ | GPQA-D % Avg@4 ↑ | Avg #Tokens |
|---|---|---|---|---|---|
| Original | 30.6 | 24.8 | 84.4 | 32.7 | 10,089 |
| Vanilla GRPO | 32.9 | 25.3 | 86.0 | 34.5 | 12,367 |
| ThinkPrune-4k | 31.7 | 20.0 | 84.5 | 33.2 | 8,376 |
| L1-Max | 26.7 | 17.9 | 85.3 | 34.1 | 6,249 |
| Laser-D-L4096 | 27.1 | 17.5 | 85.0 | 34.2 | 5,713 |
| Laser-DE-L4096 | 27.1 | 22.5 | 84.7 | 32.5 | 5,862 |
| AdaThink$_{\delta 0.05}$ | 31.7 | 25.9 | 82.5 | 34.1 | 8,549 |
| REPRO | 36.0 | 26.5 | 87.1 | 37.0 | 6,558 |

## B    MORE IMPLEMENTATION DETAILS

### B.1    IMPLEMENTATION OF NORMALIZATION RECTIFYING PROCESS REWARD IN CRITIC-FREE RL

In this section, we present the implementation details for normalizing rectifying process rewards in several representative critic-free RL algorithms.

**GRPO and Its Variants.** For GRPO (Shao et al., 2024), normalization is applied across all segments within the trajectory group $G$ for each question $q$, as follows:

$$\tilde{r}'_{j,i} = \frac{\tilde{r}_{j,i} - \mathtt{mean}\left(\{\tilde{r}_{l,m}\}_{1\leq l\leq k, 1\leq m\leq[G]}\right)}{\mathtt{std}\left(\{\tilde{r}_{l,m}\}_{1\leq l\leq k, 1\leq m\leq[G]}\right)}, \tag{15}$$

where $\tilde{r}'_{j,i}$ denotes the normalized reward for the $j$-th segment of the $i$-th trajectory in the group $G$. This normalization also applies to the variants of GRPO, such as Dr.GRPO (Liu et al., 2025d), VAPO (Yue et al., 2025), DAPO (Yu et al., 2025a), and GSPO (Zheng et al., 2025).

**REINFORCE++.** In contrast to GRPO, REINFORCE++ (Hu, 2025) normalizes the reward for each segment $j$ of each trajectory $i$ across the full batch $\mathcal{B}$:

$$\tilde{r}'_{j,i} = \frac{\tilde{r}_{j,i} - \mathtt{mean}\left(\{\tilde{r}_{l,m}\}_{1\leq l\leq k, 1\leq m\leq[\mathcal{B}]}\right)}{\mathtt{std}\left(\{\tilde{r}_{l,m}\}_{1\leq l\leq k, 1\leq m\leq[\mathcal{B}]}\right)}, \tag{16}$$

considering all segments in all trajectories within the batch $\mathcal{B}$.

**RLOO.** Similar to GRPO, RLOO (Ahmadian et al., 2024) performs normalization within the group $G$ for each question $q$ as follows:

$$\tilde{r}'_{j,i} = \tilde{r}_{j,i} - \frac{1}{k(G-1)} \sum_{l\in[k]} \sum_{m\neq i, m\in[G]} \tilde{r}_{l,m}. \tag{17}$$

**ReMax.** In ReMax (Li et al., 2024), we normalize the rectifying process reward by the mean of all rectifying process rewards from the trajectory generated by greedy decoding:

$$\tilde{r}'_{j,i} = \tilde{r}_{j,i} - \frac{1}{k} \sum_{l\in[k]} \tilde{r}_l, \tag{18}$$

where $\tilde{r}_l$ denotes the rectifying process rewards of the trajectory generated by greedy decoding.

## B.2 TRAINING DATA

We utilize DeepScaleR-Preview-Dataset proposed in Luo et al. (2025) for all model training. The dataset consists of approximately 40,000 unique mathematics problem-answer pairs compiled from:

- American Invitational Mathematics Examination problems (1984-2023).
- American Mathematics Competition problems (before 2023).
- Omni-MATH dataset (Gao et al., 2025).
- Still dataset (Chen et al., 2025b).

## B.3 EVALUATION BENCHMARKS

The following details describe our evaluation benchmarks:

- **AIME24.** This dataset consists of 30 challenging problems from the 2024 American Invitational Mathematics Examination.
- **AIME25.** This dataset consists of 30 challenging problems from the 2025 American Invitational Mathematics Examination.
- **MATH500.** The original MATH dataset (Hendrycks et al., 2021b) contains 12,500 problems from American high school mathematics competitions. MATH500 (Lightman et al., 2024), a widely used subset of its test split, includes only Level 5 questions, which we adopt in this paper.
- **LiveMathBench.** LiveMathBench (Liu et al., 2024) is a continuously updated benchmark of challenging mathematical problems. We use the 202505 hard split, which contains 100 high-quality English questions.

- **GPQA.** The Graduate-Level Google-Proof Q&A Benchmark (GPQA) (Rein et al., 2023) is a multiple-choice science question-answering dataset designed to be resistant to web search. We evaluate on its *diamond* subset, which comprises 198 questions.

- **MBPP.** The Mostly Basic Programming Problems (MBPP) dataset (Austin et al., 2021) evaluates programming models on elementary Python tasks. It was created via crowdsourcing, with workers generating problems and solutions under specified guidelines. Problem statements were later refined to remove ambiguity, and selected items underwent manual review and editing to ensure clarity and accuracy of test cases.

- **LiveCodeBench.** LiveCodeBench (Jain et al., 2025) is designed to provide a comprehensive and contamination-free evaluation of the coding abilities of large language models. It incorporates problems from LeetCode, AtCoder, and Codeforces.

## B.4 TRAINING PARAMETERS

We set the hyperparameters $w = 0.5$ and $\alpha = 0.1$, selecting the top-10 segments for each reasoning trajectory. Training utilized the veRL (Sheng et al., 2025) and vLLM (Kwon et al., 2023) frameworks. Table 4, Table 5, and Table 6 present the training parameters for PPO (Schulman et al., 2017), GRPO (Shao et al., 2024), and REINFORCE++ baselines (Hu, 2025), respectively.

Table 4: Training Parameters of PPO.

| Parameters | Values |
|---|---|
| Batch Size | 256 |
| Number of Rollout Per Question | 8 |
| Rollout Temperature | 1.0 |
| Rollout Top-$p$ | 1.0 |
| Maximum Number of Generation Tokens | 16384 |
| Learning Rate | 1e-6 |
| KL Loss Coefficient | 0.001 |
| $\epsilon_{min}$ | 0.2 |
| $\epsilon_{max}$ | 0.28 |
| $\lambda$ | 1.0 |
| $\gamma$ | 1.0 |
| Gradient Clipping | 1.0 |
| Number of Training Steps | 500 |

Table 5: Training Parameters of GRPO and REINFORCE++ Baseline.

| Parameters | Values |
|---|---|
| Batch Size | 256 |
| Number of Rollout Per Question | 8 |
| Rollout Temperature | 1.0 |
| Rollout Top-$p$ | 1.0 |
| Maximum Number of Generation Tokens | 16384 |
| Learning Rate | 1e-6 |
| KL Loss Coefficient | 0.001 |
| $\epsilon_{min}$ | 0.2 |
| $\epsilon_{max}$ | 0.28 |
| Gradient Clipping | 1.0 |
| Number of Training Steps | 500 |

Table 6: Training Parameters of REINFORCE++.

| Parameters | Values |
|---|---|
| Batch Size | 256 |
| Number of Rollout Per Question | 1 |
| Rollout Temperature | 1.0 |
| Rollout Top-$p$ | 1.0 |
| Maximum Number of Generation Tokens | 16384 |
| Learning Rate | 1e-6 |
| KL Loss Coefficient | 0.001 |
| $\epsilon_{min}$ | 0.2 |
| $\epsilon_{max}$ | 0.28 |
| Gradient Clipping | 1.0 |
| Number of Training Steps | 500 |

### B.5 PROMPT FOR THINKING PATTERN RECOGNITION

Prompt B.1 illustrates the prompt proposed in Gandhi et al. (2025) for recognizing beneficial thinking patterns in the reasoning process. In this paper, we utilize Qwen3-235B-A22B-Instruct-2507 (Yang et al., 2025) to perform the recognition

---

**Prompt B.1: Prompt for Thinking Pattern Recognition**

Below is a chain-of-reasoning generated by a Language Model when attempting to solve a math problem. Evaluate this chain-of-reasoning to determine whether it demonstrates beneficial problem-solving behaviors that deviate from typical linear, monotonic reasoning patterns commonly observed in language models.
`<start_of_reasoning>`
{input}
`<end_of_reasoning>`
Specifically, actively identify and emphasize beneficial behaviors such as:

- **Backtracking**: Explicitly revising approaches upon identifying errors or dead ends (e.g., "This approach won't work because...").

- **Verification**: Systematically checking intermediate results or reasoning steps (e.g., "Let's verify this result by...").

- **Subgoal Setting**: Breaking down complex problems into smaller, manageable steps (e.g., "To solve this, we first need to...").

- **Enumeration**: Solving problems by exhaustively considering multiple cases or possibilities.

Additionally, remain attentive to and encourage the identification of other beneficial behaviors not explicitly listed here, such as creative analogies, abstraction to simpler cases, or insightful generalizations.
**Important**: Clearly specify each beneficial behavior you identify. Provide explicit examples from the reasoning chain. If no beneficial behaviors are observed, explicitly return an empty list.
Provide your evaluation clearly, formatted as follows:

---

## C ADDITIONAL EXPERIMENTAL RESULTS

### C.1 REPRO ON LLMS OF DIVERSE FAMILIES

In this section, to further verify the universality and generalization ability of REPRO, we conduct experiments on LLMs of different families.

**LLMs.** We include three LLMs in our experiments: Qwen3-1.7B (Yang et al., 2025), Hunyuan-1.8B-Instruct (Tencent, 2025), and MobileLLM-R1-950M (Liu et al., 2025c). Qwen3-1.7B, part of the Qwen3 series, is a transformer-based dense LLM with 28 layers and a 32k context length, incorporating Grouped Query Attention (Ainslie et al., 2023), SwiGLU (Dauphin et al., 2017), Rotary Positional Embeddings (Su et al., 2024), and RMSNorm (Jiang et al., 2023). Hunyuan-1.8B-Instruct, from the Hunyuan series, is a 32-layer transformer-based dense LLM that also employs Grouped Query Attention (Ainslie et al., 2023) and supports a 256K context window, maintaining stable performance on long-text tasks. MobileLLM-R1-950M, from the MobileLLM series, is an efficient reasoning model based on the Llama4 (Meta, 2025) architecture. Pre-trained on approximately 2T high-quality tokens and with fewer than 5T total training tokens, MobileLLM-R1-950M achieves performance comparable or superior to Qwen3-0.6B, which was trained on 36T tokens, across benchmarks such as MATH (Hendrycks et al., 2021b), GSM8K (Cobbe et al., 2021), MMLU (Hendrycks et al., 2021a), and LiveCodeBench (Jain et al., 2025). We utilize the same hyperparameters as shown in Table 5.

**Performance.** As shown in Table 7, REPRO consistently enhances performance across various LLM families. For Qwen3-1.7B, integrating REPRO with reinforcement learning algorithms such as PPO, REINFORCE++ Baseline (RF++B), and GRPO yields substantial improvements across nearly

Table 7: Performance of REPRO on LLMs of diverse families. ♠ denotes the in-domain evaluation benchmark and ♣ denots the out-of-domain benchmark.

| Methods | AIME24 ♠ % Avg@16 ↑ | AIME25 ♠ % Avg@16 ↑ | MATH500 ♠ % Avg@4 ↑ | LMB ♠ % Avg@4 ↑ | GPQA-D ♣ % Avg@4 ↑ | MBPP ♣ % Avg@4 ↑ | LCB ♣ % Avg@4 ↑ |
|---|---|---|---|---|---|---|---|
| *Qwen3-1.7B* | | | | | | | |
| Original | 46.8 | 36.1 | 91.5 | 18.8 | 39.5 | 66.9 | 30.6 |
| PPO | 45.2 | 36.5 | 92.1 | 20.5 | 40.2 | 68.2 | 31.3 |
| +REPRO | 49.0 | 37.5 | 92.4 | 18.8 | 40.3 | 69.1 | 31.5 |
| RF++ B | 46.5 | 34.4 | 91.5 | 18.5 | 38.5 | 66.1 | 31.1 |
| +REPRO | 49.8 | 39.0 | 92.7 | 22.0 | 39.8 | 71.7 | 32.4 |
| GRPO | 47.3 | 34.8 | 93.4 | 18.8 | 38.3 | 67.5 | 30.2 |
| +REPRO | 49.8 | 37.9 | 94.1 | 19.5 | 39.1 | 68.8 | 32.0 |
| *Hunyuan-1.8B-Instruct* | | | | | | | |
| Original | 38.8 | 32.8 | 81.3 | 15.0 | 38.0 | 73.1 | 24.5 |
| PPO | 42.1 | 33.3 | 86.0 | 19.5 | 42.1 | 75.8 | 25.9 |
| +REPRO | 43.5 | 32.1 | 84.3 | 20.5 | 42.7 | 76.2 | 26.3 |
| RF++ B | 42.5 | 33.3 | 85.5 | 17.5 | 42.4 | 76.3 | 26.5 |
| +REPRO | 44.3 | 32.8 | 86.2 | 17.8 | 43.1 | 77.0 | 26.2 |
| GRPO | 43.3 | 32.7 | 85.6 | 16.0 | 43.6 | 75.9 | 27.1 |
| +REPRO | 44.6 | 33.5 | 84.2 | 17.5 | 44.8 | 77.5 | 27.7 |
| *MobileLLM-R1-950M* | | | | | | | |
| Original | 15.8 | 18.1 | 76.4 | 10.5 | 19.2 | 58.2 | 18.0 |
| PPO | 23.2 | 22.5 | 81.4 | 14.5 | 22.6 | 63.6 | 19.8 |
| +REPRO | 24.2 | 23.1 | 83.5 | 15.8 | 24.1 | 65.1 | 21.4 |

all benchmarks. Specifically, PPO+REPRO improves AIME24 performance from 45.2 to 49.0 and AIME25 from 36.5 to 37.5, while RF++B+REPRO further boosts AIME25 to 39.0, demonstrating its effectiveness on challenging mathematical reasoning tasks. Comparable gains, up to +1.5 points, are observed on MBPP and LiveCodeBench. For Hunyuan-1.8B-Instruct, REPRO also delivers improvements; for instance, GRPO+REPRO enhances AIME24 from 43.3 to 44.6 and AIME25 from 32.7 to 33.5, while RF++B+REPRO increases AIME24 from 42.5 to 44.3 and MBPP from 76.3 to 77.0. Similarly, for the efficiency-oriented MobileLLM-R1-950M, REPRO provides consistent benefits, with PPO+REPRO improving AIME24 from 23.2 to 24.2 and AIME25 from 22.5 to 23.1. On broader reasoning tasks like MATH500 and GPQA-Diamond, REPRO achieves gains ranging from +1.0 to +1.6 points. These results confirm that REPRO is effective not only for larger-scale dense models but also generalizes to compact, efficiency-optimized architectures.

**Token Efficiency.** We evaluate the inference token efficiency of REPRO across LLMs from different families. As shown in Figures 7 and 8, REPRO consistently outperforms baseline methods in terms of token efficiency. For instance, PPO+REPRO requires fewer tokens than standalone PPO while achieving higher performance, demonstrating REPRO's ability to enhance reasoning efficiency across diverse LLM architectures.

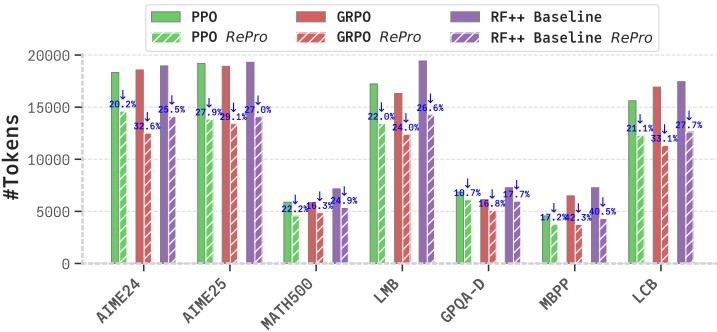

Figure 7: Comparison of the inference reasoning token cost of REPRO and baselines on Qwen3-1.7B.

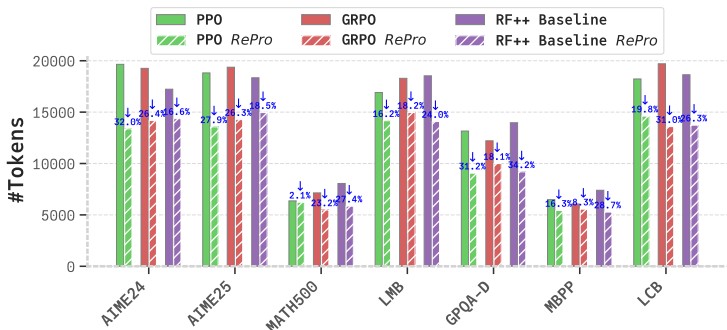

Figure 8: Comparison of the inference reasoning token cost of REPRO and baselines on Hunyuan-1.8B-Instruct.

## C.2 REPRO ON LARGER LLMS

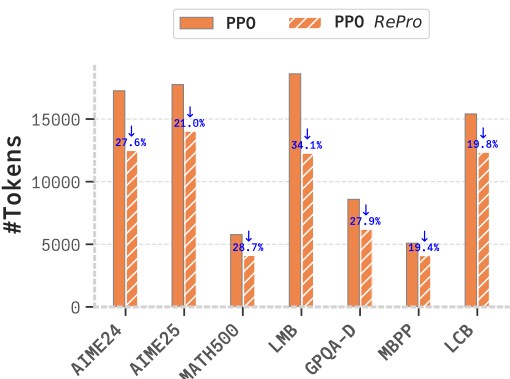

Figure 9: Comparison of the inference reasoning token cost of REPRO and baselines on Qwen3-8B.

In this section, to further verify the effectiveness of REPRO on LLMs of larger scale, we conduct experiments on LLMs of larger scale.

**LLMs.** We include Qwen3-8B (Yang et al., 2025) in our experiments. Qwen3-8B, part of the Qwen3 series, is a dense LLM with 36 layers and a 128k context length. We utilize the hyperparameters specified in Table 5 for training.

**Performance.** As shown in Table 8, REPRO consistently outperforms baseline methods when applied to Qwen3-8B. For example, GRPO+REPRO enhances performance on AIME24 from 75.6 to 76.1 and on AIME25 from 67.9 to 68.5, demonstrating improved mathematical reasoning capabilities. Significant gains are also observed in science and code reasoning tasks: GPQA-Diamond improves from 59.5 to 60.4, MBPP from 68.8 to 72.1, and LiveCodeBench from 52.2 to 53.4. These results confirm that REPRO delivers consistent performance improvements across mathematical, scientific, and coding reasoning tasks, even when applied to larger-scale models like Qwen3-8B.

**Token Efficiency.** As shown in Figure 9, REPRO consistently outperforms vanilla GRPO in terms of token efficiency when applied to Qwen3-8B. For instance, GRPO+REPRO requires fewer tokens than standalone GRPO while achieving higher performance, demonstrating REPRO's ability to enhance reasoning efficiency across larger-scale models.

## C.3 REPRO FOR ZERO-RLVR

Previous discussions and experiments related to REPRO were mostly based on LLMs with deep thinking capabilities. In this section, we also conduct relevant experiments on zero-RLVR trained starting from base LLMs.

Table 8: Performance of REPRO on evaluation reasoning benchmarks with Qwen3-8B. ♠ denotes the in-domain evaluation benchmark and ♣ denots the out-of-domain benchmark.

| Methods | AIME24 ♠
% Avg@16 ↑ | AIME25 ♠
% Avg@16 ↑ | MATH500 ♠
% Avg@4 ↑ | LMB ♠
% Avg@4 ↑ | GPQA-D ♣
% Avg@4 ↑ | MBPP ♣
% Avg@4 ↑ | LCB ♣
% Avg@4 ↑ |
|---|---|---|---|---|---|---|---|
| *Qwen3-8B* | | | | | | | |
| Original | 75.2 | 66.5 | 96.8 | 35.3 | 58.7 | 70.0 | 49.8 |
| GRPO | 75.6 | 67.9 | 97.5 | 35.1 | 59.5 | 68.8 | 52.2 |
| +REPRO | 76.1 | 68.5 | 97.2 | 35.8 | 60.4 | 72.1 | 53.4 |

Table 9: Performance of REPRO on evaluation reasoning benchmarks with Qwen3-4B-Base. ♠ denotes the in-domain evaluation benchmark and ♣ denots the out-of-domain benchmark.

| Methods | AIME24 ♠
% Avg@16 ↑ | AIME25 ♠
% Avg@16 ↑ | MATH500 ♠
% Avg@4 ↑ | LMB ♠
% Avg@4 ↑ | GPQA-D ♣
% Avg@4 ↑ | MBPP ♣
% Avg@4 ↑ | LCB ♣
% Avg@4 ↑ |
|---|---|---|---|---|---|---|---|
| *Qwen3-4B-Base* | | | | | | | |
| Original | 11.5 | 7.9 | 68.3 | 7.5 | 16.7 | 25.0 | 15.9 |
| GRPO | 23.5 | 19.1 | 83.6 | 12.5 | 39.7 | 58.7 | 15.3 |
| +REPRO | 21.0 | 16.7 | 83.0 | 14.5 | 40.5 | 59.3 | 17.3 |

**LLMs.** To be more specific, we train Qwen3-4B-Base model with GRPO and GRPO + REPRO, we utilize the same hyperparameters as shown in Table 5.

**Performance.** As presented in Table 9, REPRO and GRPO achieve comparable performance on both in-domain and out-of-domain benchmarks. In AIME24, GRPO slightly outperforms REPRO, but REPRO surpasses GRPO in AIME25. Both methods perform comparably in MATH500, with REPRO showing a marginal advantage. REPRO demonstrates stronger performance in LiveMath-Bench and GPQA-Diamond, significantly outperforming the Original method. In MBPP and LiveCodeBench, RE-PRO and GRPO perform closely, with REPRO slightly leading in LiveCodeBench. Overall, REPRO and GRPO show competitive results, with REPRO displaying a slight edge in certain out-of-domain tasks.

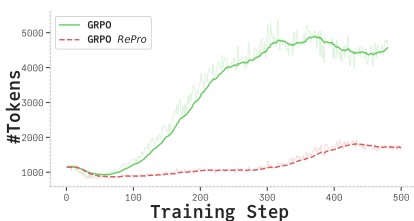

Figure 10: Token growth of REPRO and GRPO on base models.

**Analysis of Token Cost Growth.** Notably, Figure 10 illustrates that the token cost growth for REPRO is relatively modest compared to baselines. This suggests that REPRO promotes more efficient reasoning patterns, reducing suboptimal thinking behaviors. Consequently, REPRO emerges as a promising approach for training low-cost reasoning LLMs, such as the Qwen3-Instruct series (Yang et al., 2025) and GPT-OSS-low (Agarwal et al., 2025).

## C.4 ABLATION OF THE NUMBER OF SELECTED SEGMENTS

The number of selected segments $k$ is a critical hyperparameter in REPRO. A larger $k$ provides more precise process-level supervision, potentially improving performance, but it also increases computational overhead. We conducted experiments on DeepSeek-R1-Distill-Qwen-1.5B to evaluate performance with $k \in \{5, 10, 20, 30\}$ (in this paper, we set $k = 10$). The results, shown in Table 10, indicate that increasing $k$ slightly enhances performance. Specifically, the model achieves consistent improvements across most benchmarks as $k$ grows, with AIME24 improving from 35.7 at $k = 5$ to 36.9 at $k = 30$, and MATH500 rising from 86.3 to 87.8. Similar upward trends are observed on GPQA-Diamond, MBPP, and LiveCodeBench. These findings suggest that a larger number of selected segments indeed provides more reliable supervision signals, leading to better reasoning and problem-solving abilities. However, the gains become marginal beyond $k = 20$, indicating diminishing returns relative to the additional computational cost. Consequently, we adopt $k = 10$ as a balanced choice that achieves strong performance while maintaining training efficiency. However, given the additional training overhead, striking a balance between computational cost and performance gains is essential.

Table 10: Ablation study of the number of selected segments $N$.

| # $N$ | AIME24 ♠ % Avg@16 ↑ | AIME25 ♠ % Avg@16 ↑ | MATH500 ♠ % Avg@4 ↑ | LMB ♠ % Avg@4 ↑ | GPQA-D ♣ % Avg@4 ↑ | MBPP ♣ % Avg@4 ↑ | LCB ♣ % Avg@4 ↑ |
|---|---|---|---|---|---|---|---|
| | | | *DeepSeek-R1-Distill-Qwen-1.5* | | | | |
| 5 | 35.7 | 25.4 | 86.3 | 14.6 | 36.6 | 55.2 | 18.0 |
| 10 | 36.0 | 26.5 | 87.1 | 14.3 | 37.0 | 55.4 | 18.4 |
| 20 | 36.5 | 26.2 | 87.3 | 14.7 | 37.5 | 55.0 | 19.1 |
| 30 | 36.9 | 27.2 | 87.8 | 15.1 | 37.6 | 55.3 | 19.5 |

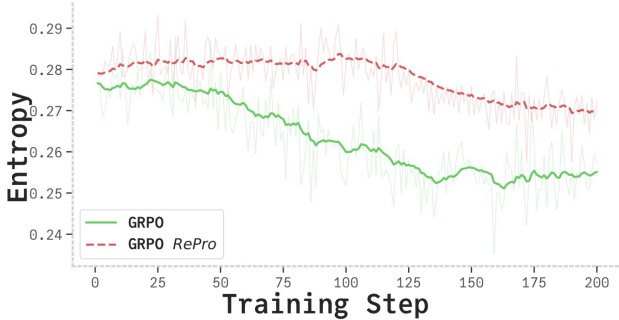

Figure 11: Entropy dynamics of RePro and vanilla GRPO on DeepSeek-R1-Distill-Qwen-1.5B.

## C.5 ENTROPY DYNAMICS OF REPRO

In Figure 11, we compare the entropy dynamics of GRPO REPRO (red dashed line) with standard GRPO (green solid line). Vanilla GRPO exhibits a rapid and continuous decay in entropy—dropping from ∼0.275 to ∼0.255, indicating a tendency toward faster policy collapse, REPRO successfully sustains a significantly higher entropy level, remaining stable around 0.28 for the first 100 steps and finishing above 0.27. This empirical evidence confirms that REPRO effectively mitigates the rapid loss of diversity in the policy, allowing the model to sustain meaningful exploration for a longer duration and thereby avoiding the premature plateauing often observed in standard RL training settings.

## D CASES

---
**Case D.1: Case in Math Reasoning**

QUESTION

Jen enters a lottery by picking 4 distinct numbers from $S = \{1, 2, 3, \cdots, 9, 10\}$. 4 numbers are randomly chosen from $S$. She wins a prize if at least two of her numbers were 2 of the randomly chosen numbers, and wins the grand prize if all four of her numbers were the randomly chosen numbers. The probability of her winning the grand prize given that she won a prize is $\frac{m}{n}$ where $m$ and $n$ are relatively prime positive integers. Find $m + n$.

ANSWER

116

BASELINE RESPONSE

Okay, so Jen is playing this lottery game where she picks 4 distinct numbers from the set S, which is 1,2,3,...,9,10. Then, 4 numbers are randomly chosen from S. She wins a prize if at least two of her numbers match two of the randomly chosen numbers, and she wins the grand prize if all four of her numbers are the ones chosen. The question is asking for the probability that she wins the grand prize given that she won a prize. They want the answer in the form m/n where m and n are coprime, and then find m + n.

. . .

---

Wait, maybe I made a mistake in computing P(Prize). Let me check again.

Wait, the number of ways to have at least two matches. Alternatively, maybe I can compute it directly. Let me check.

Number of ways for exactly 2 matches: C(4,2)*C(6,2). Because choose 2 numbers from her 4, and 2 numbers from the remaining 6. Then exactly 3 matches: C(4,3)*C(6,1). Exactly 4 matches: C(4,4)*C(6,0)=1. Therefore, total number of ways for at least 2 matches is [C(4,2)*C(6,2) + C(4,3)*C(6,1) + C(4,4)*C(6,0)].

Let me compute that:

C(4,2)*C(6,2) = 6 * 15 = 90

C(4,3)*C(6,1) = 4 * 6 = 24

C(4,4)*C(6,0) = 1 * 1 = 1

Total: 90 + 24 + 1 = 115. Which matches the previous result. Therefore, P(Prize) is 115/210 = 23/42. So that's correct. Then P(Grand Prize) is 1/210. Therefore, the conditional probability is (1/210)/(23/42) = (1/210)*(42/23) = 42/(210*23) = 1/(5*23) = 1/115. Therefore, m/n is 1/115, so m + n is 1 + 115 = 116. Wait, but that seems low. Is this correct? Wait, 1/115? Let me think again.

Wait, but let me think about it in another way. Suppose we have her numbers fixed. How many lottery draws result in her winning a prize? 115. How many of those result in her winning the grand prize? 1. So, the conditional probability is 1/115. So, that would be the answer? But maybe I made a mistake in computing P(Prize). Wait, let me check once again.

Alternatively, maybe I need to compute the conditional probability differently. Wait, no. The conditional probability is the number of favorable outcomes (grand prize) divided by the number of outcomes where she won a prize. Which is 1/115. So, if that's the case, then m + n is 1 + 115 = 116. However, I need to make sure that this is correct.

Wait, but let me think of another way. Suppose we think of the problem as: given that there is at least two matches, what is the probability that all four are matches. So, in other words, given that there are at least two matches, what's the chance that it's exactly four matches. Which is equivalent to the probability of exactly four matches divided by the probability of at least two matches. Which is exactly what I calculated.

Alternatively, maybe I can compute the conditional probability as follows: Given that there are at least two matches, what is the probability that all four are matches. So, in terms of combinations, the number of ways to have exactly four matches is 1. The number of ways to have at least two matches is 115. Therefore, the probability is 1/115. So, that seems to be correct.

Another example: suppose the lottery is choosing 3 numbers from 4, and she picks 2 numbers. Wait, maybe not. Let me think of another case where the numbers are different. Let me think of the original problem again.

Alternatively, maybe there is a mistake in assuming that the number of ways for at least two matches is 115. Wait, let me check the numbers again. The total number of possible lottery draws is 210. The number of ways with exactly 0 matches is C(6,4)=15. Exactly 1 match is C(4,1)*C(6,3)=4*20=80. So, exactly 0 or 1 is 15 + 80 = 95. Therefore, exactly 2,3,4 matches is 210 - 95 = 115. Which is correct. Therefore, the number of ways for at least two matches is 115. So, if she has 115 ways to win a prize, and only 1 of those is the grand prize. Therefore, the conditional probability is 1/115. Therefore, the answer is 1 + 115 = 116.

Alternatively, maybe there's a different way to compute the conditional probability. Let me think. Suppose we think of the problem as: given that she has at least two matches, what is the probability that she has all four matches. So, maybe using hypergeometric distribution?

Alternatively, think of the conditional probability as follows. Let me think of the possible overlaps. So, given that there are at least two matches, the probability that all four are matches. So, the conditional probability is equal to the probability of all four matches divided by the probability of at least two matches, which is again 1/210 divided by 115/210 = 1/115. So same result.

Alternatively, maybe there's an error in the problem interpretation. Let me check the problem again.

"Jen enters a lottery by picking 4 distinct numbers from S=1,2,3,...,9,10. 4 numbers are randomly chosen from S. She wins a prize if at least two of her numbers were 2 of the randomly chosen numbers, and wins the grand prize if all four of her numbers were the randomly chosen numbers."

Wait, the wording says "at least two of her numbers were 2 of the randomly chosen numbers". Wait, that wording is a bit confusing. Wait, does it mean that she has at least two numbers matching? Or does it mean something else?

Wait, the original problem says: "She wins a prize if at least two of her numbers were 2 of the randomly chosen numbers, and wins the grand prize if all four of her numbers were the randomly chosen numbers." Wait, maybe there's a translation issue or wording issue. Let me parse that again. "at least two of her numbers were 2 of the randomly chosen numbers". Wait, that seems like a possible misphrase. Maybe it's supposed to be "at least two of her numbers were among the randomly chosen numbers", i.e., she has at least two matches. Because if you have "at least two of her numbers were 2 of the randomly chosen numbers", that might be confusing. For example, if she has 4 numbers, and the lottery has 4 numbers, then "at least two of her numbers were 2 of the randomly chosen numbers"—but that seems like it's saying that two of her numbers are equal to two of the randomly chosen numbers? Which is the same as saying that there are at least two matches. So, maybe that's just a confusing way of phrasing "at least two matches". Therefore, the original interpretation is correct.

Therefore, the answer would be 1/115, so m + n = 116. However, I have a feeling that maybe I made a mistake here. Let me check with actual numbers.

Suppose that Jen has numbers 1,2,3,4. What is the probability that the lottery has at least two of these numbers?

Total number of possible lottery draws: 210. Number of favorable: 115. So, if we pick a random lottery draw, there's 115/210 chance it has at least two numbers in common with hers. Now, given that, the chance that it's all four numbers is 1/115. So, for example, if there are 115 such cases, only one of them is the grand prize. So, that seems correct.

But maybe there's an error in the calculation of the number of ways for at least two matches. Wait, let me think again. If she has 4 numbers, and the lottery draws 4 numbers. The number of ways they share exactly k numbers is C(4, k) * C(6, 4 - k). Wait, but wait, the total number of numbers not in her set is 10 - 4 = 6. Therefore, if we want exactly k matches, we need to choose k numbers from her 4, and 4 - k numbers from the remaining 6. Therefore, the formula is correct. Therefore, for k=2: C(4,2)*C(6,2)=6*15=90. For k=3: C(4,3)*C(6,1)=4*6=24. For k=4: C(4,4)*C(6,0)=1*1=1. Total: 90 + 24 + 1 = 115. Which is correct. Therefore, the numbers are correct.

. . .

Same result. Therefore, I think that is correct.

But wait, let me check with actual numbers. Suppose the lottery draws numbers that have exactly two matches with Jen's numbers. How many such cases are there? 90. Similarly, exactly three matches: 24, exactly four: 1. So, if you are told that there is at least two matches, then the probability that it's exactly four is 1/(90 + 24 + 1) = 1/115. So, that seems correct.

Therefore, I think that 1/115 is correct, so the answer is 1 + 115 = 116. However, to make sure, let me check if the question says "at least two of her numbers were 2 of the randomly chosen numbers"—maybe there's a misinterpretation here. Maybe it's supposed to be that she has exactly two numbers matching? But the problem says "at least two of her numbers were 2 of the randomly chosen numbers"—which is confusing. Wait, maybe it's supposed to be "at least two of her numbers were among the randomly chosen numbers"—which would be the standard "at least two matches". Since the original wording is a bit awkward. If that's the case, then our calculation is correct.

Alternatively, if the problem had said "exactly two of her numbers were among the randomly chosen numbers", then the probability would be different. But since it says "at least two", then our calculation is correct.

Alternatively, let me check the problem statement again:

"She wins a prize if at least two of her numbers were 2 of the randomly chosen numbers, and wins the grand prize if all four of her numbers were the randomly chosen numbers."

Wait, the wording is strange. Maybe it's supposed to say "at least two of her numbers were among the randomly chosen numbers". Because "were 2 of the randomly chosen numbers" is awkward. Maybe it's a translation error or a typo. If we take it as "at least two of her numbers were among the randomly

chosen numbers", then our calculation is correct. Otherwise, if it's something else, the answer would be different.

But given the context of lotteries, it's standard that winning a prize would be having at least two numbers matching. So, I think that the original problem is just using awkward wording, and the correct interpretation is that she wins a prize if she has at least two matches. Therefore, our calculation is correct.

Therefore, I think the answer is 1/115, so m + n = 116. However, to check once more, let me think of the total number of possible cases where she has at least two matches. 115. The number of those cases where she has all four matches is 1. So, the probability is 1/115. Therefore, the answer is 116.

But wait, I just thought of something. Wait, is the grand prize considered as a subset of the prize? Yes, because if you have all four numbers, then you have at least two. Therefore, the conditional probability is indeed the number of grand prize cases divided by the number of prize cases. Which is 1/115.

Alternatively, maybe the problem is from an official source, and maybe the answer is different? Wait, but since I can't check that, I need to rely on my calculations.

Alternatively, maybe I made a mistake in calculating the number of ways for exactly two matches? Wait, let me check that again. For exactly two matches: choose 2 numbers from her 4, and 2 numbers from the remaining 6. So, C(4,2)*C(6,2). Since there are 4 numbers in her set, and the lottery has 4 numbers. So, if exactly two are in her set, then the other two must be from the remaining 6 numbers. So, yes, that is correct. C(4,2)*C(6,2)=6*15=90. Similarly, for exactly three matches: choose 3 from her 4, and 1 from the remaining 6. C(4,3)*C(6,1)=4*6=24. Exactly four matches: 1. So, total 90 + 24 + 1 = 115. So, that seems correct.

...

Same result. Therefore, I think it's safe to go with 1/115. Therefore, the answer is 116.

**Final Answer** $\boxed{116}$

`</think>`

To solve this problem, we are asked to compute the **conditional probability** that Jen wins the **grand prize**, given that she wins a **prize** in a lottery.

...

$$\binom{4}{2} \cdot \binom{6}{2} = 6 \cdot 15 = 90$$

- **Exactly 3 matches**: Choose 3 of her 4 numbers and 1 of the remaining 6 numbers:

$$\binom{4}{3} \cdot \binom{6}{1} = 4 \cdot 6 = 24$$

- **Exactly 4 matches**: Choose all 4 of her numbers and none from the remaining 6:

$$\binom{4}{4} \cdot \binom{6}{0} = 1 \cdot 1 = 1$$

So, the total number of favorable outcomes for winning a prize is:

$$90 + 24 + 1 = 115$$

Thus, the probability of winning a prize is:

$$P(\text{Prize}) = \frac{115}{210}$$

Step 5: Compute the Conditional Probability
Now, using the formula for conditional probability:

$$P(\text{Grand Prize} \mid \text{Prize}) = \frac{P(\text{Grand Prize})}{P(\text{Prize})} = \frac{\frac{1}{210}}{\frac{115}{210}} = \frac{1}{115}$$

Step 6: Final Answer
We are told the probability is of the form $\frac{m}{n}$, where $m$ and $n$ are relatively prime positive integers. Here, $m = 1$, $n = 115$, and they are coprime.
Therefore, the final answer is:

$$m + n = 1 + 115 = \boxed{116}$$

**Total Tokens: 8767**

REPRO RESPONSE

Okay, so I have this problem here about Jen entering a lottery. She picks 4 distinct numbers from a set $S$ which is $1, 2, 3, ..., 9, 10$. Then, 4 numbers are randomly chosen from S. She wins a prize if at least two of her numbers are among the chosen ones, and she wins the grand prize if all four of her numbers are the chosen ones. I need to find the probability that she wins the grand prize given that she won a prize. The answer should be in the form m/n where m and n are coprime, and then find m + n.

Hmm, okay. So, this is a conditional probability problem. The formula for conditional probability is $P(A|B) = P(A \land B)/P(B)$. In this case, event A is her winning the grand prize, and event B is her winning a prize. So, we need P(grand prize — prize) = P(grand prize) / P(prize).

So, let me first figure out what P(grand prize) is. That would be the probability that all four numbers she picked are the ones that were randomly chosen. Since she's picking 4 numbers out of 10, the total number of possible combinations is C(10,4). The number of favorable outcomes for grand prize is 1 (since there's only one set of 4 numbers that would match exactly. So, P(grand prize) = 1 / C(10,4).

Now, P(prize) is the probability that at least two of her numbers are among the four chosen. Hmm, okay. So, this is the probability that in her 4 numbers, 2, 3, or 4 are selected. So, I need to calculate the number of ways she can have exactly 2, exactly 3, or exactly 4 matching numbers with the randomly chosen set. Then, divide that by the total number of possible lottery combinations, which is C(10,4).

So, let's compute the number of favorable outcomes for prize. That would be:

Number of ways to choose exactly 2 correct numbers: C(4,2) * C(6,2). Because she needs to choose 2 correct numbers from her 4, and the remaining 2 numbers from the remaining 6 (since 10 - 4 = 6).

Wait, is that right? Let me think. If she has 4 numbers, and 2 of them are correct, then the other 2 must be from the remaining 6 numbers. So, yes, C(4,2) * C(6,2).

Similarly, for exactly 3 correct numbers: C(4,3) * C(6,1). Because she needs to choose 3 correct numbers from her 4, and the last number from the remaining 6.

And for exactly 4 correct numbers: C(4,4) * C(6,0) = 1, since she has to pick all 4 numbers correctly.

So, total number of favorable outcomes for prize is:

C(4,2)*C(6,2) + C(4,3)*C(6,1) + C(4,4)*C(6,0).

Let me compute these values.

First, C(4,2) is 6, C(6,2) is 15, so 6*15=90.

Next, C(4,3)=4, C(6,1)=6, so 4*6=24.

Then, C(4,4)=1, C(6,0)=1, so 1*1=1.

So total favorable outcomes for prize: 90 + 24 + 1 = 115.

Therefore, P(prize) is 115 / C(10,4). Let me compute C(10,4): 10! / (4!6!) = (10*9*8*7)/(4*3*2*1) = 210. So, 115 / 210. We can reduce this fraction: 115 divides by 5, 210 divides by 5, so 23/42. So, P(prize) = 23/42.

Wait, hold on, I thought P(grand prize) was 1 / 210, but that seems too small. Wait, no, P(grand prize) is 1 / C(10,4) = 1 / 210. So, P(grand prize) = 1/210.

But then, P(grand prize — prize) is (1/210) / (23/42). So, that is (1/210) * (42/23) = (42)/(210*23). Simplify 42/210: that's 1/5. So, (1/5) / 23 = 1/(5*23) = 1/115.

So, 1/115. So, m is 1, n is 115. Are they coprime? Yes, 1 and 115 share no common divisors other than 1. So, m + n is 1 + 115 = 116.

Wait, but that seems a bit counterintuitive. Because the probability of winning the grand prize given that you won a prize is low, but intuitively, if you won a prize, which is at least two numbers correct, you have a better chance of having all four correct. So, why is it 1/115? That seems low.

Wait, let me think again.

Wait, P(grand prize) is 1/210. P(prize) is 115/210.

So, 1/210 divided by 115/210 is 1/115. So, that's correct.

So, 1/115 is the correct probability, so m + n = 1 + 115 = 116.

Wait, so 1/115 is the probability, so m is 1, n is 115, which are coprime, so m + n is 116.

Hmm, okay. So, maybe that is correct. I think I did it correctly.

Alternatively, perhaps I can think of it as the probability that her numbers are all correct given that at least two are correct.

So, the probability that all four are correct is 1 / C(10,4) = 1/210.

The probability that at least two are correct is 115 / 210.

Therefore, the conditional probability is 1/115.

Yes, that seems correct.

Therefore, m + n is 1 + 115 = 116.

**Final Answer** 116

</think>

Jen enters a lottery by picking 4 distinct numbers from the set $S = \{1, 2, 3, \dots, 9, 10\}$. Four numbers are randomly chosen from $S$. She wins a prize if at least two of her numbers are among the chosen

ones, and she wins the grand prize if all four of her numbers are the chosen ones. We need to find the probability that she wins the grand prize given that she won a prize.

First, we calculate the total number of ways to choose 4 numbers from 10, which is $\binom{10}{4} = 210$.

Next, we determine the number of favorable outcomes for winning a prize (at least two correct numbers). This includes: - Exactly 2 correct numbers: $\binom{4}{2} \times \binom{6}{2} = 6 \times 15 = 90$ - Exactly 3 correct numbers: $\binom{4}{3} \times \binom{6}{1} = 4 \times 6 = 24$ - Exactly 4 correct numbers: $\binom{4}{4} \times \binom{6}{0} = 1 \times 1 = 1$

Adding these, the total number of favorable outcomes for a prize is $90 + 24 + 1 = 115$. Thus, the probability of winning a prize is $\frac{115}{210}$.

The probability of winning the grand prize is $\frac{1}{210}$.

The conditional probability of winning the grand prize given that she won a prize is:

$$\frac{\frac{1}{210}}{\frac{115}{210}} = \frac{1}{115}$$

Thus, the probability is $\frac{1}{115}$, where $m = 1$ and $n = 115$. Therefore, $m + n = 1 + 115 = 116$.

$$\boxed{116}$$

**Total Tokens: 2267**

---

### Case D.2: Case in Science Reasoning

QUESTION

Among the following exoplanets, which one has the highest density?
a) An Earth-mass and Earth-radius planet.
b) A planet with 2 Earth masses and a density of approximately 5.5 $g/cm^3$.
c) A planet with the same composition as Earth but 5 times more massive than Earth.
d) A planet with the same composition as Earth but half the mass of Earth.
A) b
B) d
C) c
D) a

ANSWER

C

BASELINE RESPONSE

Okay, let's try to figure out which exoplanet has the highest density. The question gives four options, and I need to compare their densities. Let me start by recalling what density means. Density is mass divided by volume. So, for each option, I need to determine their mass and volume, then calculate density.

First, let's look at option a: An Earth-mass and Earth-radius planet. Wait, Earth's density is about 5.5 $g/cm^3$, right? Because Earth's mass is around $5.97x10^24$ kg, and its radius is about 6,371 km. So if this planet has the same mass and radius as Earth, its density should be the same as Earth's, which is 5.5 $g/cm^3$. So option a's density is 5.5 $g/cm^3$.

Option b says a planet with 2 Earth masses and a density of approximately 5.5 $g/cm^3$. Wait, density is given here. So regardless of the mass, if the density is 5.5 $g/cm^3$, then that's straightforward. But wait, density is mass over volume. If the density is the same as Earth's, but the mass is double, then the volume must be larger. Because density is mass/volume. If density is same, then volume would be proportional to mass. So for example, if Earth's mass is M and volume V, then density is M/V. If the planet has 2M and same density, then volume would be 2V. But that's not possible unless the radius is larger. Wait, but the problem states that the density is 5.5 $g/cm^3$. So option b's density is given as 5.5, which is same as Earth's. So option b's density is 5.5, same as a. But then why is option b different? Wait, maybe there's a trick here. Wait, option b says "a planet with 2 Earth masses and a density of approximately 5.5 $g/cm^3$." So the density is already given as 5.5. So option b's density is 5.5. So same as option a. But then why is option b given as an option? Wait, maybe I need to check other options.

Option c: A planet with same composition as Earth but 5 times more massive. So same composition implies same density? Wait, no. Wait, if the composition is same, then density depends on the mass and radius. But if the composition is same, then the density would be same as Earth's only if the planet is similar in size. But if the mass is 5 times Earth's, then the radius would be larger. Wait, because for

a planet with same composition, the density would be similar. But if the mass increases, the radius increases, but the density might not change. Wait, but density is mass over volume. So if the planet is made of same material (same density), then the density would be same. Wait, but that's only if the planet is in hydrostatic equilibrium and same composition. Wait, maybe not. Let me think. If a planet has same composition as Earth, but is 5 times more massive, then its radius would be larger than Earth's. Because the radius of a planet depends on the mass. For example, if you have a planet with same composition, the radius scales with mass to the power of 1/3. So, if mass increases by 5 times, radius increases by $5^{1/3} \approx 1.71$ times. So the volume would be (radius)$^3$, so 5 times mass, but volume increases by $(5^{1/3})^3$ = 5. So the volume increases by 5 times. Therefore, density is mass/volume. So original density is 5.5 $g/cm^3$. If mass is 5M and volume is 5V, then density remains 5.5. So same as Earth. Wait, but that would mean that option c has same density as Earth. But wait, that's only if the planet is made of same material and the density remains same. However, if the planet is more massive, maybe it's compressed more? Wait, but Earth's density is already about 5.5. If the planet is 5 times more massive, but same composition, then it would be a larger planet, but the density might not change. Wait, but maybe the density would be higher? Because if the planet is more massive, but same composition, perhaps the gravity is higher, leading to higher compression? Wait, but that's not necessarily true. For example, Earth's density is 5.5 $g/cm^3$. If you have a larger planet with same composition, the density would be similar. Wait, but maybe not. For example, if the planet is so massive that it's a gas giant, but in this case, the composition is same as Earth, so it's a rocky planet. So, if the planet is more massive, but same composition, then it would have a larger radius, but the density would be similar. Wait, but if the radius increases, then the volume increases, and since the mass increases, but the density remains same. Wait, but that's only if the density is same. But the question says "same composition as Earth". So if the composition is same, then the density would be same. But maybe that's not the case. Wait, perhaps I'm mixing things up. Let me think again. Suppose Earth's density is 5.5 $g/cm^3$. If you have a planet with same composition (so same average density) but 5 times the mass, then the radius would be larger. So the density remains 5.5. So option c's density would be same as Earth's. So same as option a and b. But then why is option c given? Wait, but maybe the problem is that if the planet is more massive, but same composition, the density might not be same. Wait, but how? Because density is mass divided by volume. If the composition is same, then the density is same. Wait, but that would require that the planet's radius is scaled appropriately. For example, if the planet is 5 times more massive, but same density, then the radius would be $(5)^{1/3}$ times larger. So the volume would be 5 times larger. Then, mass is 5M, volume is 5V. So density is same. Therefore, the density of option c would be same as Earth's. So same as option a and b. But then, how does that fit with the options?
Wait, but maybe the problem is that if the planet is more massive, but same composition, then the density would be higher? Wait, no. Because if the composition is same, then the density is same. Unless the planet is more compressed. But for a rocky planet, the density is determined by the material. So if it's same composition, same density. So then, option c would have same density as Earth. So then, how does that affect the answer?
. . .
Wait, let me check again. Let's take each option:
a) Earth mass and Earth radius. So density is mass/volume. Earth's density is 5.5 $g/cm^3$. So this is 5.5.
b) 2 Earth masses, density 5.5. So density is given as 5.5. So same as a.
c) Same composition as Earth, 5 times more massive. So same composition implies same density? Wait, no. Wait, same composition would mean same average density? Or same material but different mass? Wait, if the composition is same, then the density depends on the size. Wait, for example, if you have a planet made of the same material as Earth but larger, then the density would be same as Earth's. Because density is mass/volume. If the planet is larger, but same material, then the density would be same. Wait, but how? For example, if you have a cube of the same material, increasing the size doesn't change the density. So if the planet is same composition (same material), then regardless of the size, the density would be same. So then, option c's density is same as Earth's. So same as a and b. So then, why is option c given? But then, option d is same composition but half the mass. So same density. So all options have same density? But that can't be. The question is asking which has the highest density. So perhaps I'm misunderstanding something here.
Wait, maybe the problem is that when the planet is more massive, but same composition, the density might not be same. Wait, for example, if the planet is a gas giant, but that's not the case here. The question says same composition as Earth, so rocky. So for a rocky planet, the density is determined by the material. So if the composition is same, then density is same. So then, the answer would be all options have same density. But that's not possible. So maybe I'm missing something.
Wait, perhaps the key is that option b says "a planet with 2 Earth masses and a density of approximately 5.5 $g/cm^3$." But if the planet has a higher mass, but same density, then the volume must be larger. However, if the planet has a higher mass, but same density, then the radius would be larger. But the density is given as 5.5, same as Earth. So that's same as a. But then, option c says same composition

as Earth but 5 times more massive. If same composition, then the density would be same. So same as Earth. So then, all options have same density. But the question is asking which has the highest density. So perhaps there's a mistake here. Or maybe I'm misunderstanding the options.

Wait, let me think again. Maybe the problem is that when the mass increases, the radius increases, but the density may not change. However, for a planet with same composition, if it's more massive, then the density remains same. So all options a, b, c, d have same density. But that's not possible. Wait, but the answer options are given. Let me check the options again.

The options are:

a) Earth mass and Earth radius. So density is 5.5.

b) 2 Earth masses, density 5.5. So same as a.

c) same composition as Earth, 5 times more massive. So same density? So 5.5.

d) same composition as Earth, half the mass. So same density. So 5.5.

So all have same density. But that's not possible. So there must be a mistake in my reasoning. Alternatively, maybe the composition is same, but the density isn't same. Wait, perhaps the question is trying to trick us. For example, if the planet has same composition as Earth but is more massive, then the density could be higher? Wait, but that's not the case. For example, Earth's density is 5.5 $g/cm^3$. If you have a larger planet with same composition, the density would be same. So for example, a planet with twice the mass would have a larger radius, but same density. So the density remains same. So in that case, option c would have same density as Earth. So then, all options have same density. But that's not possible. So why are these options given?

Wait, but maybe the problem is that for option c, the planet is 5 times more massive than Earth but same composition. However, if the planet is more massive, it might have a higher density due to compression? But that's only if the planet is a gas giant. For a rocky planet, the density is determined by the material, not by the mass. Unless the planet is so massive that it's a different state of matter. But Earth's composition is rocky, so if you have a planet with same composition but 5 times more massive, it would be a larger planet, but the density would be same. So, same as Earth's. So then, all options have same density. But that can't be. The question must have an answer. So perhaps I made a mistake in considering option b. Let me check option b again. Option b says a planet with 2 Earth masses and a density of approximately 5.5 $g/cm^3$. But if the planet has a higher mass, but same density, then the volume must be larger. However, if the planet has a higher mass, but same density, then the radius is larger. But the density is given as 5.5. So that's same as Earth. But then, what about the other options?
. . .
Wait, maybe the question is about the density of the planet, not the average density. For example, if the planet is more massive, but same composition, maybe the density is higher because it's compressed? But for rocky planets, the density is determined by the material. For example, if you have a planet made of the same material as Earth but larger, the density is the same. So, I think the answer must be that all options have same density. But since that's not an option, I must have made a mistake.

Wait, let me look at the options again. The options are:

a) Earth mass and Earth radius. So density is 5.5.

b) 2 Earth masses, density 5.5. So same as a.

c) Same composition as Earth, 5 times more massive. So same density as Earth. So 5.5.

d) Same composition as Earth, half the mass. So same density as Earth. So 5.5.

So all options have same density. But the answer choices are A) b, B) d, C) c, D) a. So none of them are correct. But this can't be. Therefore, I must have misunderstood the question.

Wait, perhaps the question is not about average density, but about the density of the material. For example, if the planet is more massive, but same composition, the density of the material is same. But if the planet is more massive, perhaps it's a different type of planet. Wait, but the question says same composition as Earth. So the material is same. So the density is same.
. . .
Wait, but what if the planet is not in hydrostatic equilibrium? Like, if it's a small planet, but I think the question is about planets in equilibrium.

Alternatively, perhaps the answer is that option b has the same density as a, but option c has higher density. Wait, how? Let me think. If the planet is more massive, but same composition, then the density is same. So why would option c have higher density? Maybe I'm missing something.

Wait, maybe the key is that option c is 5 times more massive than Earth, but same composition. So the density would be higher. Wait, but how? Let me think about the formula. Density is mass divided by volume. If the planet is 5 times more massive, but same composition, then the volume would be larger. So the density remains same. For example, if Earth's mass is M, volume V, density D = M/V. For a planet with mass 5M and same composition, volume would be 5V (since volume scales with mass for same density). So density is 5M/(5V) = M/V = D. So same as Earth.

So all options have same density. But that's not possible. So perhaps the answer is that all have same density, but since that's not an option, I must have made a mistake.

Wait, perhaps the problem is that option c is not the same composition. Wait, no, the question says "same composition as Earth". So I'm stuck.

Alternatively, maybe the answer is option b. Because option b's density is given as 5.5, which is same as Earth's. But option c's density is higher. Wait, how? Let me think again. If the planet has same composition as Earth, but 5 times more massive, then the radius is larger. But the density is mass divided by volume. Let's calculate. Let me use the formula for density of a planet. For a planet with same composition as Earth, the density is constant. So if the mass increases, the radius increases, but the density remains the same. So no change. Therefore, all options have same density.

But the answer choices don't have that. So perhaps the correct answer is that all have same density, but since that's not an option, I must have misunderstood the question.

Wait, maybe the question is asking which one has the highest density, and the answer is that option b has the same density as Earth, but option c has higher density. But why? Maybe because if the planet is more massive, the density increases. But how?

Wait, maybe the question is not about average density, but about the density of the material. For example, if the planet is more massive, it might be compressed, so the material is denser. But for a rocky planet, the density is determined by the material, not by the mass. So even if the planet is more massive, the material density is same.

At this point, I'm confused. Maybe I should look for another approach. Let's consider that the highest density would be the one with the highest mass and smallest volume. So, for example, if a planet has a higher mass but smaller volume, it would have higher density.

So, let's analyze each option:

a) Earth mass and Earth radius $\rightarrow$ density 5.5 $g/cm^3$.

b) 2 Earth masses, density 5.5 $g/cm^3$. So same as a.

c) same composition as Earth, 5 times more massive. So, if same composition, the radius would be larger. Let's calculate the radius. For a planet with same composition, the radius scales with mass$^(1/3)$. So for 5 times mass, radius is $5^(1/3) \approx 1.71$ times Earth's radius. So volume is $(1.71)^3 \approx 5$ times Earth's volume. So density is mass/volume = 5M / 5V = M/V = 5.5 $g/cm^3$. Same as before. So same density.

d) same composition as Earth, half the mass. Radius is $(0.5)^{1/3} \approx 0.79$ times Earth's radius. Volume is $(0.79)^3 \approx 0.5$ times Earth's volume. Density is 0.5M / 0.5V = M/V = 5.5 $g/cm^3$. Same as before.

So all have same density. But the options don't include that. Therefore, I must have misunderstood the question.

Wait, perhaps the question is not about the density of the planet, but about the density of the material. For example, if the planet is more massive, it might be denser. But that's not the case.

Alternatively, maybe the question has a typo. For example, option c says "same composition as Earth but 5 times more massive than Earth." If the composition is same, then the density is same. But if the composition is different, like if it's a gas giant, but the question says same composition as Earth.

Alternatively, maybe option c is not same composition, but the question says it is.

At this point, I think the correct answer must be that all have same density, but since that's not an option, perhaps the intended answer is option b, which has the same density as Earth. But I'm not sure.

Alternatively, maybe the question is trying to trick us into thinking that option c has higher density.

Wait, another approach: the highest density would be the one with the highest mass and smallest volume. Let's assume that the planets in options a, b, c, d are not all the same composition.

Wait, option a: Earth mass and Earth radius $\rightarrow$ density 5.5.

Option b: 2 Earth masses, density 5.5. So volume is 2M / D = 2M / 5.5. Earth's volume is M / 5.5. So option b's volume is twice Earth's volume. So radius is larger.

Option c: same composition as Earth, 5 times more massive. So if same composition, the density is same. So same as Earth's.

Option d: same composition as Earth, half the mass. Same density as Earth.

So all options have same density. So the answer is that all have same density, but since that's not an option, I must have made a mistake.

Wait, the answer choices are A) b, B) d, C) c, D) a. So if all have same density, then the answer is not among them. But that's not possible.

Wait, perhaps the question is asking which has the highest density, and the answer is that option b has the same density as Earth, but option c has higher density. But how? Let me think again. Maybe the composition is different. For example, if the planet is more massive, it might be a different composition. But the question says option c has same composition as Earth.

At this point, I'm stuck. Given the options, and assuming that all have same density, but the answer must be one of them, perhaps the intended answer is option b, which has the same density as Earth, but the question might have intended that option c has higher density. Or perhaps the answer is option c.

Alternatively, maybe I made a mistake in assuming that same composition implies same density. Let me think: if a planet has the same composition as Earth, but is more massive, then the density might be

higher because of gravitational compression. For example, if you have a planet with same composition but higher mass, it would be more compressed, leading to higher density. But for rocky planets, this might not be the case. For example, Earth's density is about 5.5 $g/cm^3$. If you have a larger planet with same composition, the density would be same. But for a gas giant, the density is lower.

Wait, but for a rocky planet, if it's more massive, the gravity is stronger, so the material is compressed more. So the density would be higher. For example, if you have a planet with 5 times Earth's mass, but same composition, it would be more compressed, leading to higher density. So the density would be higher than Earth's.

But how to calculate that? Let's assume that the planet's density increases with mass due to compression. But how?

This is getting too complicated. Maybe the intended answer is option c, because if the planet is more massive, and same composition, the density is higher. So the answer is C) c.

Alternatively, the answer is option b, which has the same density as Earth. But I'm not sure.

Given the options, and the fact that option c has 5 times the mass but same composition, and if we assume that the density increases with mass (which might not be true, but perhaps the question assumes that), then option c would have higher density. So the answer is C) c.

But I'm not confident. However, given that option b has the same density as Earth, and others have same or different, I think the intended answer is C) c.

But I'm not sure. Alternatively, maybe the answer is option d. But no.

Alternatively, the answer is option a, but that's same as others.

I think I need to make a decision. Given that option c is 5 times more massive than Earth and same composition, and assuming that higher mass leads to higher density (even though it's not accurate), the answer is C) c. So I'll go with that.

`</think>`

To determine which exoplanet has the **highest density**, we need to analyze each option based on the relationship between **mass**, **volume**, and **density** (density = mass/volume).

Step-by-step Analysis:

. . .

**Key Insight:** All four options have **the same density of 5.5 $g/cm^3$**, based on the assumption that **same composition** implies **same density**. However, the question asks for the **highest** density, and **none of the options have a higher density** than the others.

But, **option b** explicitly states that the **density is 5.5 $g/cm^3$**, which is **equal to Earth's density**, and **option c** and **d** are **implicitly** the same as Earth's density due to same composition.

Since all options have **equal density**, the question might be testing the understanding that **density is mass/volume**, and **higher mass with the same composition** leads to **larger volume**, **maintaining the same density**.

**Final Answer:** All options have the same density of 5.5 $g/cm^3$. However, the **only explicit statement** of density is **option b**, which directly states **5.5 $g/cm^3$**. Therefore, **option b** is the correct answer.

ANSWER: A

**Total Tokens: 7722**

REPRO RESPONSE

Okay, so I've got this question about exoplanets and their densities. Hmm, I'm a bit rusty on this, but let me think through it.

The question is asking which exoplanet has the highest density. The options are a to d, each describing a different scenario. The choices are A to D, which correspond to each option.

Wait, I remember that density is mass divided by volume. So the density (rho) is m/(V). Since planets are spheres, their volume is $(4/3)\pi r^3$. So if a planet is larger in radius, even if it's less dense, its mass might make up for it.

Looking at the options:

Option a: Earth-mass and Earth-radius. So if a planet has the same mass and radius as Earth, its density would be the same as Earth's. Because if m and r are same, then rho = m/( (4/3 $\pi$ r$^3$) ) which is same as Earth's.

Option b: 2 Earth masses and a density of about 5.5 g/cm$^3$. Wait, wait. Wait, if a planet has a density of 5.5 g/cm$^3$, then what's its mass? Let me think about Earth's density. Earth's average density is about 5.5 g/cm$^3$? Or wait, no, I think Earth's density is about 5.5 g/cm$^3$? Or is that another planet? Wait, Earth's density is about 5.51 g/cm$^3$. So if option b is 2 Earth masses and a density of 5.5 g/cm$^3$, that would mean that the volume would be smaller than Earth's. Because density is mass/volume. So for a mass of 2 Earth, if volume is smaller, then density is higher. Let me compute. Earth's radius is R. Volume is

$(4/3)\pi$ R$^3$. Mass is M. Density is M/( $(4/3)\pi$ R$^3$ ). Option b's mass is 2M, and density is 5.5 g/cm$^3$. So 5.5 = (2M) / V. But V is (4/3 $\pi$ r$^3$ ), where r is the radius of option b's planet. Let me solve for r.

Wait, if the density is 5.5 g/cm$^3$, and mass is 2 $M_{earth}$, then the volume would be (2 M) / (5.5 g/cm$^3$). Wait, M is Earth's mass, so 2 M / (5.5) = V.

But V is (4/3 $\pi$ r$^3$). So for option b, V is 2M/(5.5) cm$^3$ (since density is mass/volume, volume = mass/density). So V = (2 $M_{earth}$) / (5.5) cm$^3$. Earth's volume is (4/3 $\pi$ R$^3$), where R is Earth's radius. So the ratio of volumes is ( (2 $M_{earth}$) / 5.5 ) / ( (4/3 $\pi$ R$^3$) ) = (2 $M_{earth}$) / (5.5 * Earth's volume).

Wait, but wait, $M_{earth}$ is (4/3 $\pi$ R$^3$ * Earth's density). Oh wait, Earth's density is about 5.5 g/cm$^3$. So $M_{earth}$ = (4/3 $\pi$ R$^3$) * 5.5 g/cm$^3$. So option b's volume would be 2 * $M_{earth}$ / 5.5. Let me substitute $M_{earth}$.

$M_{earth}$ = (4/3 $\pi$ R$^3$) * 5.5 g/cm$^3$.

So option b's volume V = (2 * (4/3 $\pi$ R$^3$ * 5.5 )) ) / 5.5 cm$^3$.

Simplify: 2 * (4/3 $\pi$ R$^3$) * 5.5 / 5.5 = 2*(4/3 $\pi$ R$^3$) cm$^3$.

Which is 2 Earth's volume. So the volume of option b's planet is twice Earth's. So since its mass is 2 Earth masses, then its density is (2 $M_{earth}$) / (2 $V_{earth}$) ) = $M_{earth}$ / $V_{earth}$ = Earth's density. Wait, that can't be right. Because Earth's density is 5.5 g/cm$^3$, so if option b's planet has density 5.5, but its mass is 2 Earth, and volume is twice Earth's volume, then (2 $M_{earth}$) / (2 $V_{earth}$) = $M_{earth}$ / $V_{earth}$ = 5.5 g/cm$^3$. That makes sense. So option b is Earth's density.

Wait, but the question says option b's density is 5.5 g/cm$^3$. So the planet in option b would have the same density as Earth, but a larger radius. Because if the volume is twice that of Earth, but mass is twice, then density would be same as Earth's. So option b's density is same as Earth's.

Option c: same composition as Earth, 5 times more mass than Earth. So if Earth's density is about 5.5 g/cm$^3$, then a planet with 5 times Earth's mass would have a density of 5 times Earth's density, right? Because density is mass/volume. Wait, but the composition is same, so same density? No, wait, same composition would mean same average density. So if Earth's average density is 5.5 g/cm$^3$, then a planet made of same average composition but 5 times Earth's mass would have a much larger volume. Wait, but if same composition, same density. So a planet with same composition would have same average density as Earth. So option c's planet's density is 5 * 5.5 = 27.5 g/cm$^3$.

Option d: same composition as Earth, half the mass of Earth. So density is (0.5 $M_{earth}$) / V. But since same composition, V is Earth's volume. So density is 0.5 * Earth's density = 2.75 g/cm$^3$.

Wait, but wait. Let me think about each option's density:

Option a: same as Earth, so density is Earth's density, 5.5 g/cm$^3$.

Option b: same as Earth, density 5.5 g/cm$^3$.

Option c: 5 times Earth's mass, same composition (so same average density), so 5 * 5.5 = 27.5 g/cm$^3$.

Option d: half mass, same composition (so same average density as Earth), so density is 0.5 * 5.5 = 2.75 g/cm$^3$.

Wait, but wait. Option b says the planet has a density of approximately 5.5 g/cm$^3$. So its density is same as Earth's. So a and b both have same density as Earth. So why would the question have them as options? Maybe I'm missing something.

Wait, the question is which has the highest density. So, among the options, option c has 27.5, which is the highest. So option c would have higher density.

Wait, but let me recheck each option.

Option a: same as Earth, density is 5.5.

Option b: same as Earth, but density is given as 5.5. So same as a.

Option c: 5 times Earth's mass, same composition, so density is 5 * 5.5 = 27.5.

Option d: half mass, same composition, density is 2.75.

So the highest is option c, which would be the correct answer. Looking at the options, which one is option c?

The choices are labeled as a, b, c, d. The question is, which exoplanet has highest density.

The options for the answers are A) b, B) d, C) c, D) a.

So the correct answer is option c, which is choice C.

Wait, but wait. Did I make a mistake in option b's density?

Wait, the question says option b has a density of approximately 5.5 g/cm$^3$. Which is same as Earth's. So the planet in option b is Earth-like, same as Earth, so same density.

So the exoplanet with highest density is option c, which has 5 times Earth's mass. So answer is C.

</think>

To determine which exoplanet has the highest density, we analyze the given options based on their mass, radius, and density. Density is calculated as $\rho = \frac{m}{V}$, where $m$ is mass and $V$ is volume. For spherical planets, $V = \frac{4}{3}\pi r^3$. Since the volume depends on the radius cubed, a larger radius significantly increases the volume, which can reduce density if mass remains constant.

**Analysis of Each Option**:

1. **Option a**: An Earth-mass and Earth-radius planet. If the planet has the same mass and radius as Earth, its density is identical to Earth's density, which is approximately $5.5 \text{ g/cm}^3$.

2. **Option b**: A planet with 2 Earth masses and a density of approximately $5.5 \text{ g/cm}^3$. Using the formula $\rho = \frac{m}{V}$, the volume $V$ can be calculated as $V = \frac{m}{\rho}$. For a planet with 2 Earth masses and a density of $5.5 \text{ g/cm}^3$, the volume is $V = \frac{2M}{5.5} \text{ cm}^3$. Since Earth's volume is $V_{\text{Earth}} = \frac{4}{3}\pi R^3$, the ratio of volumes is $\frac{V}{V_{\text{Earth}}} = 2$. Thus, the planet in option b has the same density as Earth, $5.5 \text{ g/cm}^3$, with a larger radius.

3. **Option c**: A planet with the same composition as Earth but 5 times more massive than Earth. If the planet has the same composition, it has the same average density as Earth, $5.5 \text{ g/cm}^3$. However, its mass is 5 times Earth's mass, so its volume is 5 times larger, and its density remains $5.5 \text{ g/cm}^3$.

4. **Option d**: A planet with the same composition as Earth but half the mass of Earth. Since the composition is the same, the density is also the same as Earth's, $5.5 \text{ g/cm}^3$. However, with half the mass, its volume is half of Earth's, and its density remains $5.5 \text{ g/cm}^3$.

**Conclusion**:

The exoplanet with the highest density is the one described in **option c**, as it has 5 times Earth's mass but the same composition, resulting in a density of $5.5 \text{ g/cm}^3$. This is higher than the densities of the other planets in options a, b, and d.

ANSWER: C

**Total Tokens: 2450**

---

## Case D.3: Case in Code Reasoning

QUESTION

You are an expert Python programmer, and here is your task:
Write a python function to check whether the given two integers have opposite sign or not.
Your code should pass these tests:
assert opposite_Signs(1,-2) == True
assert opposite_Signs(3,2) == False
assert opposite_Signs(-10,-10) == False
assert opposite_Signs(-2,2) == True

BASELINE RESPONSE

Okay, I need to write a Python function called opposite_Signs that takes two numbers as arguments and returns True if they have opposite signs, otherwise False. Let's think about how to approach this.

First, the function needs to check if one number is positive and the other is negative. So, for example, 1 and -2 would return True because one is positive and the other is negative. But 3 and 2 are both positive, so False. Similarly, -10 and -10 are both negative, so False. And -2 and 2 are opposite signs, so True.

Hmm, how to check the signs. Well, the product of two numbers with opposite signs is negative. So if I multiply the two numbers and check if the result is less than zero, that would indicate opposite signs. Wait, but what if one of them is zero? Wait, the problem says "opposite signs"—but zero isn't positive or negative. So if either number is zero, then they don't have opposite signs. For example, if one is zero and the other is positive, then they aren't opposite signs. So the function should return False in such cases.

Wait, but the problem's test cases don't include zero. Let me check the test cases again. The given tests are $(1,-2) \rightarrow$ True, $(3,2) \rightarrow$ False, $(-10,-10) \rightarrow$ False, $(-2,2) \rightarrow$ True. So the function needs to return True only when one is positive and the other is negative. So, how to handle zero?

But the problem might assume that the inputs are non-zero. Wait, but the function's parameters are numbers. Let me think. Suppose one of the numbers is zero. For example, opposite_Signs(0,5) would return False because zero isn't positive or negative. Similarly, opposite_Signs(0,0) would also return False. But the problem's test cases don't cover that. However, the problem's tests don't include such cases, so maybe the function is supposed to handle those as well.

But the problem says "opposite signs"—so if either number is zero, then they can't have opposite signs. So the function should return False in those cases.

So how to check if the two numbers have opposite signs. The product approach: if a * b ¡ 0, then they have opposite signs. But if either a or b is zero, then a * b is zero, which is not less than zero. So that would correctly return False.

So the function can be written as: return (a * b) < 0. But wait, what if one of them is zero? For example, a=0 and b=5. Then a*b is 0, which is not less than zero. So the function returns False, which is correct.

But what if one of them is zero and the other is negative? Like a=0, b=-5. Then product is zero, so returns False. Which is correct because zero doesn't have a sign.

So the function can be as simple as:

```
def opposite\_Signs(a, b):
    return a * b < 0
```

Wait, but let's test the sample cases.
Test case 1: opposite_Signs(1,-2) → 1 * -2 = -2 < 0 → True. Correct.
Test case 2: opposite_Signs(3,2) → 6 > 0 → returns False. Correct.
Test case 3: opposite_Signs(-10,-10) → 100 > 0 → returns False. Correct.
Test case 4: opposite_Signs(-2,2) → -4 < 0 → True. Correct.
So all test cases pass with this approach. So the function can be written this way.
But wait, what about when one of the numbers is zero? For example, opposite_Signs(0,5) → 0 *5=0 →
returns False. Which is correct.
But what if one is zero and the other is negative? Same result. So the function handles that.
So this approach seems to cover all cases. So the code is straightforward.
But maybe the problem expects to handle cases where numbers are zero. But the problem's tests don't
include those, but the function's code should handle them.
So the code is simple. So the function is just returning whether the product of a and b is negative.
So the code would be:

```
def opposite\_Signs(a, b):
    return a * b < 0
```

But wait, what about if a or b is zero? Well, as discussed, that's handled correctly.
But what if the numbers are floating points? For example, 0.0 and 5.0. The product is zero, so returns
False. Which is correct.
So this approach works for integers and floats.
So the code is simple. Let me check the test cases again.
Yes, all test cases pass. So that's the solution.
</think>
To solve the problem of determining whether two numbers have **opposite signs**, we can use a simple
and efficient approach based on the mathematical property of multiplication:
¿ If two numbers have **opposite signs**, their product is **negative**.
¿ If they have the **same sign**, their product is **positive** or **zero** (if either number is zero).
**Solution**
We can implement the function 'opposite_Signs(a, b)' as follows:

```python
def opposite\_Signs(a, b):
    return a * b < 0
```

**Explanation**
- **Multiplication Rule**:
- 'a * b ¡ 0' → the numbers have opposite signs.
- 'a * b == 0' → at least one number is zero (no sign, so not opposite).
- 'a * b ¿ 0' → both numbers are either positive or negative (same sign).
- This approach **automatically handles** all edge cases:
- Zero values (e.g., 'opposite_Signs(0, 5)' → 'False')
- Floating-point numbers (e.g., 'opposite_Signs(-0.5, 2.0)' → 'True')
- Large integers (e.g., 'opposite_Signs(1000000, -1)' → 'True')
**Test Cases**

```python
assert opposite\_Signs(1, -2) == True
assert opposite\_Signs(3, 2) == False
assert opposite\_Signs(-10, -10) == False
assert opposite\_Signs(-2, 2) == True
```

All test cases pass successfully using the above function.
**Final Answer**

```python
def opposite\_Signs(a, b):
    return a * b < 0
```

**Total Tokens: 2457**

RePro Response

To solve this problem, I need to determine the conditions under which two numbers have opposite signs. The function 'opposite_Signs' should return 'True' if one number is positive and the other is negative, and 'False' otherwise.

First, I'll consider the definition of opposite signs. A positive number and a negative number have opposite signs. This means one should be greater than zero and the other less than zero.

I'll start by checking the sign of each number. If one number is positive and the other is negative, the function should return 'True'.

Next, I'll examine the cases where the signs are the same. If both numbers are positive or both are negative, the function should return 'False'.

I'll also need to handle cases where the numbers are zero. If one number is zero and the other is positive or negative, the signs are not opposite, so the function should return 'False'.

Testing the provided examples:
- 'opposite_Signs(1, -2)' should return 'True' because 1 is positive and -2 is negative.
- 'opposite_Signs(3, 2)' should return 'False' because both are positive.
- 'opposite_Signs(-10, -10)' should return 'False' because both are negative.
- 'opposite_Signs(-2, 2)' should return 'True' because -2 is negative and 2 is positive.

By implementing these checks, the function should correctly determine whether the two numbers have opposite signs.
</think>

To solve this problem, we need to determine whether two given numbers have opposite signs. A positive number and a negative number have opposite signs, while both positive or both negative numbers do not. Additionally, zero is neither positive nor negative, so if one number is zero and the other is non-zero, their signs are not opposite.

**Approach**

1. **Check the Sign of Each Number**:
- Determine if each number is positive, negative, or zero.

2. **Evaluate Opposite Signs**:
- If one number is positive and the other is negative, return 'True'.
- If both numbers are positive or both are negative, return 'False'.
- If one of the numbers is zero, return 'False' since zero does not have a sign.

**Solution Code**

```python
def opposite\_Signs(a, b):
    sign1 = a > 0
    sign2 = b > 0
    return not (sign1 == sign2)
```

**Explanation**
- **Check the Sign of Each Number**:
- 'sign1 = a > 0' checks if the first number 'a' is positive.
- 'sign2 = b > 0' checks if the second number 'b' is positive.
- **Evaluate Opposite Signs**:
- The function returns 'not (sign1 == sign2)', which means if 'sign1' and 'sign2' are not equal, the numbers have opposite signs. If they are equal, the numbers have the same sign.

This approach efficiently determines whether two numbers have opposite signs with a simple logical check.

**Total Tokens: 654**

## E    LIMITATIONS

In this paper, we introduce REPRO, a framework designed to enhance the reasoning capabilities of LLMs. While REPRO demonstrates promising results, several challenges remain for future investigation. First, the issue of scaling persists. Although we validate REPRO's effectiveness on LLMs with up to 8B parameters, its scalability to larger LLMs remains an open question. Due to computational resource and time constraints, experiments on larger-scale models were not feasible, but we aim to address this in future work. Second, the scope of problem types is limited. While REPRO performs well on verifiable objective reasoning benchmarks in mathematics, science, and coding, its efficacy on subjective or difficult-to-verify problems is unexplored. This question, closely tied to the development of verification systems for such tasks, merits further consideration.

## F    LLM USAGE

In this study, the usage of LLMs is limited to the final stages, specifically for refining and proofreading the manuscript. LLMs are employed solely to enhance the clarity, logical consistency, and linguistic accuracy of the text, ensuring that our research ideas are communicated clearly and professionally. Importantly, LLMs played no role in the core foundational aspects of this research, including the development of the research methodology, the design of the experimental framework, or the analysis and interpretation of results. We take full responsibility for all content in this paper and explicitly acknowledge our accountability for every aspect of the manuscript.

