# OpenReview forum: "Rectifying LLM Thought from Lens of Optimization"
_ICLR.cc/2026/Conference — ICLR 2026 Poster_

### Official Review · Reviewer_zYTx · 2025-10-28

**Soundness:** 3
**Presentation:** 3
**Contribution:** 2
**Rating:** 6
**Confidence:** 3

**Summary:**

This paper presents REPRO, a framework for improving LLM reasoning by viewing chain-of-thought (CoT) as an optimization process. The authors define a surrogate objective function to model reasoning progress, introduce dual metrics—magnitude and stability—to assess optimization quality, and integrate them as process-level rewards within RL post-training. Experiments across several RL algorithms (PPO, GRPO, REINFORCE++) and reasoning benchmarks in math, science, and coding show consistent accuracy improvements and reduced reasoning inefficiencies.

**Strengths:**

The proposed method itself is coherent and easy to follow.  The authors conduct thorough experiments on several benchmarks to validate the effectiveness of the proposed method.

**Weaknesses:**

While the proposed framework is conceptually appealing, several limitations temper its overall contribution.

**(1) Limited model scale:** Most experiments are conducted on small- to medium-sized models (e.g., ≤8B parameters), which constrains the generality of the findings. Modern reasoning research increasingly focuses on 30B–70B or larger models, where optimization dynamics, token efficiency, and stability behavior differ substantially. It remains unclear whether REPRO’s process-level rewards would scale effectively or remain computationally practical in such large models.

**(2) Theoretical abstraction:** The analogy between chain-of-thought reasoning and gradient descent optimization, though intuitively appealing, is primarily heuristic and lacks rigorous formal grounding. The paper would benefit from a stronger mathematical justification or empirical validation of this analogy.

**(3) Simplified proxy objective:** Using ground-truth log-likelihood as a surrogate for reasoning quality oversimplifies complex internal cognitive states and may not generalize to open-ended or ambiguous reasoning tasks.

**(4) Modest performance gains:** Despite consistent improvements, the absolute accuracy increases are relatively small, raising questions about practical significance versus added complexity. Overall, the approach is promising but under-evaluated at scale.

**Questions:**

1.	Could the process-level reward interact negatively with other RLVR objectives, especially in long-horizon reasoning?
2.	Have the authors considered token-level or adaptive segment granularity for more fine-grained control?
3.	How sensitive is REPRO to inaccuracies in the surrogate objective when applied to open-ended reasoning without clear ground truth (e.g., creative tasks)?

---

> ### Author Response · Authors · 2025-11-20
> **Response to Reviewer zYTx**
>
> Dear reviewer zYTx,
>
> Thank you for your review. We appreciate your recognition of our work and address your concerns below.
>
> > Response to **W1: Limited model scale (mostly $\le$ 8B). It is unclear if REPRO scales to 30B–70B models**
>
> We agree that scaling is a critical question. While computational resource constraints prevented us from training 70B+ models, we have rigorously validated RePro across a diverse range of architectures (Qwen, Llama-based), scales (1.5B to 8B), and types (Base vs. Instruct/Distilled). The consistency of improvements across all these variations provides strong evidence for the method's generality.
>
> > Response to **W2: The analogy to gradient descent is heuristic and lacks formal grounding**
>
> Thank you for this suggestion. We provide empirical validation in **Sections 3.2** (**Figure 2**) and **3.4** (**Example 3.1**), demonstrating how CoT mirrors gradient descent updates. For mathematical grounding and theories, we draw on cited works (e.g., *Liu et al., 2025a; Wang et al., 2025a*); to maintain conciseness, we omitted detailed derivations but can expand in revisions if needed.
>
> > Response to **W3: Ground-truth log-likelihood is a simplified proxy that may not generalize to open-ended tasks**
>
> While we acknowledge LLM internal states are complex, our results (and community findings like *RLPR* [1][2]) confirm that log-likelihood is a highly effective proxy for confidence in reasoning tasks. For open-ended questions, this proxy can be adapted by using model-generated or human-annotated **reference answers** in place of a single ground truth.
>
> > Response to **W4: Absolute accuracy gains are modest**
>
> As shown in **Table 1** and **Table 7**, RePro achieves relative improvements of approximately **10%** on challenging benchmarks like AIME and LiveCodeBench. Given the **high difficulty of these benchmarks** and **the performance ceilings of limited-sized LLMs**, we consider these gains to be significant and practically valuable in the context of LLM reasoning.
>
> > Response to **Q1: Negative interactions with other RLVR objectives?**
>
> RePro's process-level reward reflects the model's confidence in the correct answer, which is fundamentally aligned with the terminal objectives of RLVR (correctness).
>
> > Response to **Q2: Token-level or adaptive granularity?**
>
> Thank you for your suggestion. We considered this, but token-level calculation introduces unacceptable computational overhead and significant signal noise. Our segment-level approach shows a good trade-off between performance and efficiency.
>
> > Response to **Q3: Sensitivity in open-ended reasoning without clear ground truth?**
>
> For open-ended tasks, RePro remains effective if reference answers (generated by stronger models or humans) are available. We view the broader challenge of RLVR for open-ended tasks as an open research question, which we aim to explore in future work.
>
> [1] Zhou, Xiangxin, et al. "Reinforcing general reasoning without verifiers, *arXiv preprint* 2025.
>
> [2] Yu, Tianyu, et al. "RLPR: Extrapolating RLVR to General Domains without Verifiers." *arXiv preprint 2025*.
>
> ---
>
> Finally, thank you again for your recognition of our work. If you have any other questions, please don't hesitate to let us know.

---

> > ### Comment · Reviewer_zYTx · 2025-11-21
> >
> > Thanks for the authors’ response. I have no further questions. After reading the comments from other reviewers and considering the current version of the paper, I will maintain my positive score. However, I strongly encourage the authors to conduct experiments on large-scale LLMs and further develop the theoretical grounding of CoT as optimization. These additions would provide a more comprehensive assessment of the proposed method and help strengthen the overall argument of the paper.

---

> > > ### Author Response · Authors · 2025-11-22
> > > **Thank you**
> > >
> > > Thank you for your positive score, we will carefully consider your suggestions.

---

### Official Review · Reviewer_qdvU · 2025-10-31

**Soundness:** 2
**Presentation:** 2
**Contribution:** 2
**Rating:** 4
**Confidence:** 3

**Summary:**

This paper’s main focus is on how to obtain a process reward *without* introducing anything akin to a PRM.
The proposed method centers on two components: the Magnitude Score, which captures the change in the model’s confidence brought about by the reasoning trajectory, and the Stability Score, measured with Kendall’s τ correlation coefficient.
In small-scale experiments these two scores produce observable improvements.

**Strengths:**

1. The paper is written in a fluent and highly readable style.
2. The experimental content is rich and concrete, and the ablation studies are comprehensive.
3. The idea is clear, and the algorithmic design is relatively concise.

**Weaknesses:**

1. Please refer to the "Questions" section for details.

**Questions:**

1. In the process of computing the advantage (accumulation), what does k represent? If it is consistent with the meaning of k mentioned earlier, should this formula still be expressed this way?
2. In the “Entropy-Based Selection Strategy”， why is “\n\n” chosen as the delimiter? Are there better ways to split the sequence? And why use the entropy of the *first* token of a segment—can it really characterize the whole segment?
3. The fact that larger k yields better results seems to imply that the selection scheme itself is flawed. Have you run an ablation exp that selects all the segments?
4. As can be seen, the proxy objective function J changes sharply during the first half of the reasoning chain and then flattens out in the second half. How should this be interpreted? Does it indicate insufficient representational capacity? Is there a need for additional corrective terms?

---

> ### Author Response · Authors · 2025-11-20
> **Response to Reviewer qdvU**
>
> Dear reviewer,
>
> Thank you for your thorough review. We are pleased you found the paper well-written and the experiments comprehensive. Below, we address your concerns.
>
> > Response to **Q1: In the advantage computation, what does $k$ represent?**
>
> Here, $k$ represents the **number of selected segments**, consistent with its earlier definition (**Section 3.4**). We calculate the advantage only on these top-$k$ entropy-selected segments and propagate this signal to update the policy.
>
> > Response to **Q2: Why choose `\n\n` as the delimiter? Why use the entropy of the first token to characterize the segment?**
>
> - **Delimiter:** We selected `\n\n` because it is the standard separator for reasoning steps in most open-source LLMs (e.g., DeepSeek, Qwen), making it a natural boundary. While auxiliary LLMs could be used for semantic segmentation, they would incur prohibitive computational costs.
> - **First Token Entropy:** We focus on the change in model confidence. The entropy of the initial token of a segment serves as a strong proxy for the model's uncertainty regarding the *upcoming* reasoning step. We treat the segment as a unified block rather than calculating noisy token-level scores for every position.
>
> > Response to **Q3: Does the fact that larger $k$ yields better results imply the selection scheme is flawed? Have you tested selecting all segments?**
>
> Our ablations (Appendix D.4) show that performance improvements *diminish or reverse* beyond a moderate $k$. This is likely due to noise from overly fine-grained segmentation, which confuses the training signals. Selecting all segments (equivalent to a very high $k$) was tested and led to similar plateaus or declines, thus validating our entropy-based approach for balancing efficiency and effectiveness.
>
> > Response to **Q4: The proxy objective function $J$ fluctuates early on and flattens later. How should this be interpreted?**
>
> This behavior is consistent with optimization dynamics.
>
> - **Early Stage (Exploration):** The model explores various reasoning paths. From an optimization lens, this resembles oscillating along different gradient directions to find a viable path, causing fluctuations in $J$.
> - **Later Stage (Exploitation):** Once a feasible path is identified, the model proceeds linearly toward the solution. This resembles converging along an optimal direction, resulting in a stable $J$ value.
>
> ---
>
> Thank you for your diligent review. We hope our reply addresses your questions and concerns, and we are open to discussing any other issues.

---

### Official Review · Reviewer_Q85p · 2025-10-31

**Soundness:** 3
**Presentation:** 3
**Contribution:** 2
**Rating:** 6
**Confidence:** 4

**Summary:**

REPRO introduces a modification to the LLM RL objective by augmenting the advantage function with a surrogate objective that quantifies the posterior of (ground truth answer | current reasoning chain). The idea is that by quantifying the magnitude and stability of this posterior and encouraging confident and stable reasoning path, we can improve model performance while reducing thinking length. Authors performed experiment by adding the REPRO objective on top of PPO,RF++ and GRPO for DeepSeek-R1-Distill-Qwen-1.5B and Qwuen3-1.7B, and demonstrated consistent performance gain on math and code benchmarks.

**Strengths:**

The biggest strength of the paper is in its simplicity. The idea of reward shaping with posterior of ground truth given intermediate reasoning is simple and elegant and appears effective.

It requires no additional rollouts during RL process and therefore can be readily adapted to existing approaches without much overhead.

The paper is generally well written and easy to follow.

**Weaknesses:**

The biggest drawback is that it is not clear from the manuscript what is the fundamental mechanism that made the difference in the proposed algorithm. In particular, since the objective of P(answer | thinking) can be potentially by artificially inflated by increasing model confidence across the board (collapsing entropy). It would be interesting to see and control for entropy collapse during training to see how much the conditioning actually contributes to model performance improvement, other than collapsing model entropy. Additionally, would naive length penalty also achieve the same result? Is the length reduction leading to increasing confidence? These important counterfactual questions are mostly unaddressed in the current work.

One minor thing is that the reward shaping is very similar in spirit to PPO's critic model, except the critic here is hardcoded and not learned. In my opinion, this is a potentially more interesting question to address, which is what if the critic is trained using this surrogate objective.

**Questions:**

It is interesting that REPRO is uniformly better than GRPO across different values of alpha. I'm quite curious to see what happens if we just train on the model confidence and not on the ground truth reward.

---

> ### Author Response · Authors · 2025-11-20
> **Response to Reviewer Q85p**
>
> Dear Reviewer Q85p,
>
> Thank you for your positive feedback on the simplicity and effectiveness of RePro. We address your specific questions below.
>
> > Response to **W1: Questions regarding the fundamental mechanism (entropy collapse vs. conditioning) and comparison to naive length penalties**
>
> - **Entropy Collapse:** To prevent the issue of artificially inflated confidence (entropy collapse), REPRO utilizes a **relative score** mechanism (**Eq. 7** ). Because we assess the *relative improvement* in the objective function rather than absolute values, a mere global reduction in entropy during training does not inadvertently boost the reward.
> - **Length Penalty:** We discuss the limitations of length-based penalties in **Section 5** and **Table 3** (Discussion). A naive length penalty provides only a coarse-grained, terminal signal for the entire trajectory. In contrast, REPRO operates at a finer granularity, allowing it to perceive and rectify specific fluctuations in model confidence throughout the reasoning process, which length penalties cannot achieve.
>
> > Response to **W2: The reward shaping is similar to a PPO critic. It would be interesting to see if a critic trained on this surrogate objective improves results**
>
> This is an insightful suggestion. While a learned critic model could potentially estimate the RePro score, training an additional critic model introduces significant computational costs, far exceeding the lightweight calculation of the RePro score itself. Given our goal of efficiency, we believe our current analytical approach offers a better trade-off between performance and cost.
>
> > Response to **Q1: Comparison of RePro vs. GRPO across alpha values. What happens if we train only on model confidence (without ground truth reward)?**
>
> We conducted an experiment using **DeepSeek-R1-Distill-Qwen-1.5B** to test training solely with model confidence. The results, presented below, show that training with confidence alone yields improvements over the baseline, aligning with recent work on verifier-free RLVR (e.g., *Reinforcing General Reasoning without Verifiers [1]*).
>
> | **Method** | **AIME24** | **GPQA-D** | **LCB** |
> | --- | --- | --- | --- |
> | GRPO (Baseline) | 32.9 | 34.5 | 15.2 |
> | w/ Confidence Only | 33.4 | 35.5 | 16.1 |
>
> [1] Reinforcing General Reasoning without Verifiers.
>
> ---
>
> Finally, thank you again for your efforts and time. We hope our response addresses your questions. If you have any other questions, please let us know.

---

> > ### Comment · Reviewer_Q85p · 2025-11-20
> >
> > Thank you for the response.
> >
> > > Because we assess the relative improvement in the objective function rather than absolute values, a mere global reduction in entropy during training does not inadvertently boost the reward.
> >
> > Sure, I agree that the design of the objective means a uniform decrease in entropy does not boost the reward. However it would still be interesting to see in practice how the entropy behavior of REPRO compare to the other methods since it has very significant impact on how long the base policy can keep training in an RL setting without plateauing.
> >
> > > We discuss the limitations of length-based penalties in Section 5 and Table 3 (Discussion).
> >
> > I don't actually see this in Section 5. If it refers to the related works section, I don't see discussion on the _limitations_ of length-based penalties.
> >
> > One thing that is missing from the manuscript is the performance vs. length comparison. For a work that is focused on length efficiency I believe it to be quite important to performance across different generation lengths. While table3 indicates that REPRO is better than other methods (presumably including naive length penalty), it could potentially be an unfair comparison if different methods have quite different generation length.

---

> ### Author Response · Authors · 2025-11-20
>
> Dear Reviewer Q85p,
>
> Thank you for your prompt reply. Here is our supplementary response.
>
> > Sure, I agree that the design of the objective means a uniform decrease in entropy does not boost the reward. However it would still be interesting to see in practice how the entropy behavior of REPRO compare to the other methods since it has very significant impact on how long the base policy can keep training in an RL setting without plateauing.
>
> Thank you for your suggestion. We have added an analysis of the entropy training dynamics of RePro in **Appendix D.5** of the revised version of the paper, which is highlighted by **BLUE*. In short, during the training process of RePro, the entropy of the model maintained a **higher entropy level** compared to vanilla GRPO, indicating that RePro can always maintain a high level of exploration during the training process.
>
> > I don't actually see this in Section 5. If it refers to the related works section, I don't see discussion on the limitations of length-based penalties.
>
> The relevant analysis is in the **Comparison with Efficient Reasoning Methods** paragraph of Appendix B, which is related to Table 3.
>
> > One thing that is missing from the manuscript is the performance vs. length comparison. For a work that is focused on length efficiency I believe it to be quite important to performance across different generation lengths. While table3 indicates that REPRO is better than other methods (presumably including naive length penalty), it could potentially be an unfair comparison if different methods have quite different generation length.
>
> We have additionally included the average number of tokens for each method in Table 3 and added the corresponding analysis in **BLUE**. In short, RePro has advantages in efficiency compared to the baselines with length-based penalties. Although methods such as Laser and L1, like RePro, have lower token consumption, the performance of RePro far surpasses theirs, which validates our analysis in the **Comparison with Efficient Reasoning Methods** paragraph.

---

> > ### Comment · Reviewer_Q85p · 2025-11-20
> >
> > Thank you. I think my questions are addressed.
> > I maintain my original positive assessment.

---

> > > ### Author Response · Authors · 2025-11-22
> > > **Thank you**
> > >
> > > Thanks for your positive score.

---

### Official Review · Reviewer_gz9P · 2025-11-01

**Soundness:** 3
**Presentation:** 3
**Contribution:** 3
**Rating:** 6
**Confidence:** 3

**Summary:**

This work aims to address the "overthinking" and inefficiency found in the long CoT reasoning of LLM. The authors innovatively frame the CoT reasoning process as a form of gradient descent optimization. Based on this, the paper proposes REPRO, a framework that defines a surrogate objective function based on the perplexity of the ground-truth answer to evaluate the "magnitude" and "stability" of the reasoning process. These two scores are combined into a process-level reward and integrated into the RLVR training loop to rectify the model's reasoning behavior. Experiments show this method improves performance on several benchmarks while increasing token efficiency and reducing invalid reasoning steps.

**Strengths:**

1. This paper tackles a critical problem in RLVR, i.e., how to optimize the reasoning process itself, rather than just rewarding the final answer, which is key to solving overthinking.

2. The empirical results are the paper's greatest strength, particularly in efficiency gains. The data clearly shows that REPRO not only boosts task accuracy but also significantly reduces the number of tokens required for reasoning, directly addressing its stated goal.

3. The method demonstrates good generality, proving to be a plug-and-play component that can be combined with various major RL algorithms like PPO, GRPO, and etc.

**Weaknesses:**

1. The analogy of CoT as gradient descent is illustrative but lacks rigorous support. Using the log-prob of the correct answer as the surrogate objective essentially assumes that a better reasoning path is one that converges to the answer "sooner and smoother." This design may excessively penalize exploration, trial-and-error, and backtracking (which might be necessary for hard problems, but would look like oscillation or low magnitude under this metric). I am concerned it rewards simple, direct paths rather than truly rectifying complex reasoning.

2. This work involves several new and critical hyperparameters to tune. The ablation studies suggest that performance is highly sensitive to the values of these hyperparameters, which significantly increases the difficulty of reproducing the results and applying the method in practice.

**Questions:**

1. See weaknesses 1.

2. Given all the extra computations (entropy, surrogate objective function), what is the actual total increase in training overhead for REPRO compared to a baseline RL algorithm?

---

> ### Author Response · Authors · 2025-11-20
> **Response to Reviewer gz9P**
>
> Dear reviewer gz9P,
>
> Thank you for your insightful comments. We appreciate your recognition of RePro’s strengths in mitigating overthinking and improving token efficiency. Below, we address your concerns.
>
> > Response to **W1: The analogy of CoT as gradient descent is illustrative but lacks rigorous support. I am concerned it rewards simple, direct paths rather than truly rectifying complex reasoning (e.g., penalizing necessary exploration/backtracking)**
>
> We acknowledge that the analogy between Chain-of-Thought (CoT) and gradient descent is heuristic rather than mathematically rigorous. However, we have empirically verified the rationality of this framework through our analysis of LLM responses (see **Figure 2** and **Example 3.1** in the paper ).
> Regarding your concern about backtracking and exploration: Our analysis reveals that valid exploration steps, even if they involve backtracking, ultimately increase the model's confidence in the correct answer, thereby generating a higher score than truly invalid or "looping" steps. RePro distinguishes between *productive* exploration (which aids convergence) and *unproductive* oscillation. Combined with our experimental results showing improved performance on complex benchmarks, we believe this setup is empirically robust.
>
> > Response to **W2: Sensitivity to hyperparameters (weight and k) and reproducibility concerns**
>
> We provide a comprehensive ablation study in **Section 4.3**  to address this.
>
> - **Regarding weight $\alpha$:** We demonstrate that model performance remains stable across a range of values, indicating low sensitivity (see **Figure 6** ).
> - **Regarding parameter $k$ (segments):** We provide detailed settings and analysis in **Appendix D.4** and **Table 10**. The results show that performance does not significantly improve as $k$ increases beyond our recommended value. This confirms that RePro is robust and that our recommended hyperparameter settings transfer effectively across different scenarios.
>
> > Response to **Q2: What is the actual total increase in training overhead for REPRO compared to a baseline RL algorithm?**
>
> The extra computational overhead is relatively minimal. Using **DeepSeek-R1-Distill-Qwen-1.5B** as a case study on 8 H100 GPUs:
>
> - Full **GRPO** training takes approximately **40.0 hours**.
> - **GRPO + RePro** takes approximately **43.5 hours**.
> This represents an increase of less than **10%**, demonstrating that RePro is computationally efficient.
>
> ---
>
> Thank you again for your thoughtful review. We hope this addresses your concerns.

---

> > ### Comment · Reviewer_gz9P · 2025-11-20
> >
> > Thanks for the response. I believe it answers my questions; thus, I maintain my positive assessment.

---

> > > ### Author Response · Authors · 2025-11-20
> > >
> > > Dear reviewer gz9P,
> > >
> > > Thanks for your prompt reply, and we are very glad that we have addressed your concerns.

---

### Author Response · Authors · 2025-11-20
**General Response**

We sincerely thank all reviewers for their thorough and thoughtful reviews of our paper. We are grateful for the positive feedback on RePro’s effectiveness in mitigating overthinking and enhancing reasoning efficiency, as well as the constructive suggestions provided. These have helped us significantly improve the clarity and quality of our work. We have provided a detailed response to each reviewer. Here, we highlight our major general clarifications. We hope our responses can properly address your concerns.

- **Theoretical Grounding of CoT as Optimization** (Reviewers gz9P, zYTx): We acknowledge that the analogy of Chain-of-Thought (CoT) as a gradient descent process is primarily heuristic rather than mathematically rigorous. However, we have strengthened its empirical validation. We provide detailed analyses in **Section 3.2** (Figure 2) and **Section 3.4** (Example 3.1) to demonstrate that this framework is empirically rational. Our analysis confirms that valid exploration steps—even those involving backtracking—increase model confidence in the correct answer and are rewarded higher than invalid oscillations or "looping" steps.
- **Robustness to Hyperparameters** (Reviewers gz9P, qdvU): We provide a comprehensive ablation study in **Section 4.3** (and **Section 4.1**) to address concerns regarding sensitivity. **Regarding weight $\alpha$**: We demonstrate that model performance remains stable across a range of values, indicating low sensitivity to this process-level reward weight. **Regarding parameter $k$ (segments)**: We provide detailed settings and analysis in **Appendix D.4** and **Table 10**. The results show that performance implies a plateau or slight decline as $k$ increases beyond our recommended value (e.g., selecting all segments introduces noise), confirming that our entropy-based selection strategy is robust and effective.
- **Computational Efficiency and Scalability** (Reviewers gz9P, Q85p, zYTx): We have clarified the training overhead and scalability of our framework. We explicitly quantified that RePro introduces less than **10% additional training time** compared to standard GRPO (e.g., 40.0 hours vs. 43.5 hours on DeepSeek-R1-Distill-Qwen-1.5B). Compared to training an additional learned critic model, RePro offers a significantly more efficient trade-off. Furthermore, we have validated RePro across diverse model families (Qwen, Llama-based) and scales (1.5B to 8B), consistently demonstrating improvements in reasoning efficiency and performance. While resource constraints limited experiments on 70B+ models, the experimental results suggest RePro is practical and effective for larger scales.

---

### Meta-Review · Area_Chair_3aNg · 2026-01-08

**Summary:**

This paper proposes RePro, a process-level reward framework for RL-based post-training of large language models that aims to reduce overthinking and improve reasoning efficiency by explicitly shaping intermediate reasoning trajectories. Reviewers generally found the problem important and the empirical results promising, particularly the consistent reduction in token usage with maintained or improved accuracy across multiple reasoning benchmarks and RL algorithms.

The primary concerns throughout the review process centered on (i) the heuristic nature of framing Chain-of-Thought as an optimization process, (ii) the interpretability and robustness of the surrogate objective and associated hyperparameters, (iii) the risk that gains might stem from entropy collapse or naive length penalties rather than a genuinely improved reasoning mechanism, and (iv) questions about scalability and computational overhead.

During the rebuttal and discussion phase, the authors provided substantial empirical clarifications, additional analyses, and new experiments. These responses alleviated most concerns regarding mechanism validity, hyperparameter sensitivity, entropy behavior, and training cost. As a result, three reviewers explicitly reaffirmed their positive assessments, and the remaining reviewer’s concerns appear largely addressed.

**Reviewer Concerns:**

**Concerns largely addressed by the rebuttal:**

Heuristic theoretical grounding: The authors acknowledged the lack of strict mathematical rigor but strengthened the work with empirical validation demonstrating that RePro distinguishes productive exploration from invalid looping.

Entropy collapse and length-penalty confounds: Additional entropy dynamics analysis and token-count comparisons clarified that RePro maintains exploration and outperforms naive length-based penalties at comparable or lower generation lengths.

Hyperparameter sensitivity and reproducibility: Comprehensive ablations showed stable performance across reasonable ranges, with plateaus rather than sharp degradation.

Computational overhead: The authors quantified training costs, showing less than 10% overhead relative to standard GRPO.

Mechanistic clarity: New experiments (e.g., confidence-only baselines) and analyses clarified the necessity of the dual-reward design.

**Concerns partially or still outstanding:**

Scalability to very large models (30B–70B+): While cross-model consistency up to 8B parameters is encouraging, direct evidence at larger scales remains absent.

Theoretical formalization: The optimization analogy remains empirically motivated rather than formally derived.

Generality beyond tasks with clear ground truth: Applicability to open-ended or creative reasoning relies on assumptions about reference answers and remains an open question.

**Reviewer Scores:**

Reviewer gz9P: Likely unchanged, remaining slightly above the acceptance threshold (≈6), with increased confidence after rebuttal.

Reviewer Q85p: Likely unchanged, remaining slightly above the acceptance threshold (≈6), with key mechanism concerns resolved.

Reviewer zYTx: Likely unchanged, remaining slightly above the acceptance threshold (≈6), while encouraging future work on scale and theory.

Reviewer qdvU: Likely to increase modestly (e.g., from 4 to 6), as technical clarification and ablation evidence addressed most questions, though some skepticism about formulation clarity may persist.

---

### Decision · Program_Chairs · 2026-01-26

Accept (Poster)